# CROSS-MODAL TRANSFER THROUGH TIME FOR SENSOR-BASED HUMAN ACTIVITY RECOGNITION

## ABSTRACT

Cross-modal knowledge transfer between time-series sensors remains a critical challenge for robust Human Activity Recognition (HAR) systems. Effective cross-modal transfer exploits knowledge from one modality to train models for a completely unlabeled target modality—a problem setting we refer to as Unsupervised Modality Adaptation (UMA). Existing methods typically compress continuous-time data samples into single latent vectors during alignment, limiting their ability to transfer temporal information through real-world temporal distortions. To address this, we introduce Cross-modal Transfer Through Time (C3T), which preserves fine-grained temporal information during alignment to handle dynamic sensor data better. C3T achieves this by aligning a set of temporal latent vectors across sensing modalities. Our extensive experiments on various camera+IMU datasets demonstrate that C3T outperforms existing methods in UMA by over 8% in accuracy and shows superior robustness to temporal distortions such as time-shift, misalignment, and dilation. Our findings suggest that C3T has significant potential for developing generalizable models for time-series sensor data, opening new avenues for various multimodal applications.

## 1 INTRODUCTION

Unsupervised human activity recognition (HAR) across different sensing modalities remains a significant challenge in machine learning. A limitation of existing cross-modal feature alignment methods is that they encode an entire time-series sequence into a single latent vector, hindering the transfer of temporal information (Moon et al., 2022; Girdhar et al., 2023; Gong et al., 2023). This compression is especially problematic when training and testing on real-world continuous sensor data, such as wearable inertial data, where the same action may occur with significant temporal variations (different speeds, starting points, or durations). We introduce Cross-modal Transfer Through Time (C3T), a novel approach that preserves local temporal information during cross-modal alignment, enabling more effective knowledge transfer (Figure 1). Our experiments show C3T outperforms conventional contrastive alignment approaches and demonstrates robustness to common temporal distortions in cross-modal transfer from RGB videos to inertial sensors.

Inertial Measurement Units (IMUs), which typically provide 3-axis acceleration and 3-axis gyroscopic information on a wearable device, emerge as strong candidates for understanding human motion in a nonintrusive fashion. Smartwatches, phones, earbuds, and other wearables have enabled the seamless integration of IMUs into daily life (Mollyn et al., 2023). Unfortunately, IMUs remain underutilized in current machine-learning approaches due to several challenges, including: (1) the scarcity of labeled IMU data stemming from the difficulty in interpreting raw sensor readings, and (2) the challenge of defining precise activity boundaries given the continuous time-series nature of IMU data.

Beyond IMUs, various sensing modalities are gaining popularity in wearables (e.g., surface electrocardiogram, electromyography) and ambient monitoring systems (e.g., WiFi signals, Radar). Researchers have developed numerous AI methods for HAR using these modalities; however, these methods lack the generalization capability seen in visual and textual models, which benefit from Internet-scale data. As new sensing modalities emerge, a critical challenge is *how to effectively transfer knowledge from data-rich modalities to new sensor modalities without requiring extensive labeling effort.*

A promising solution is cross-modal transfer, where knowledge from a well-labeled source modality is transferred to an unlabeled target modality (Niu et al., 2020). However, current cross-modality and

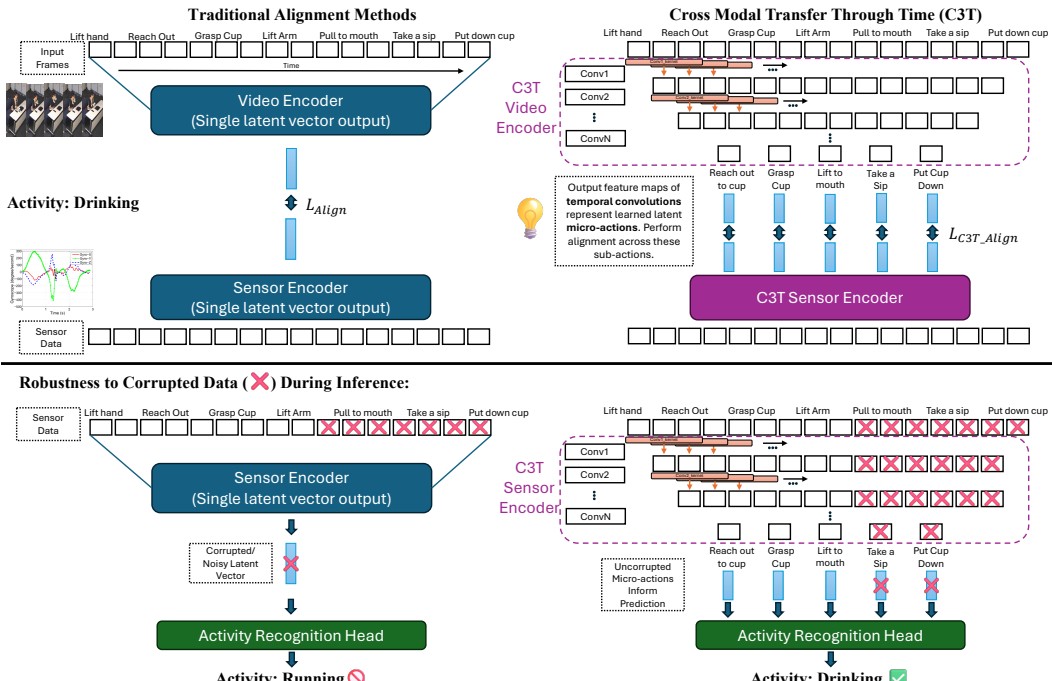

Figure 1: **Motivation for C3T:** Conventional alignment methods collapse an entire video sequence into a single vector. C3T performs alignment across local fine-grained temporal features, improving cross-modal transfer performance (Table 1) and robustness (Table 2).

missing-modality techniques are semi-supervised with some labeled examples from each modality during training (An et al., 2021; Woo et al., 2023; Garcia et al., 2018; Wang et al., 2020; Nugroho et al., 2023; Yang et al., 2022). This creates a significant barrier to adopting new sensing technologies, as each application would require costly re-labeling efforts, and inhibits real-time unsupervised transfer as described in Figure 2. To address these practical challenges, we explore a setting where the target modality is completely unlabeled during training. We refer to this as *Unsupervised Modality Adaptation* (UMA), akin to Unsupervised Domain Adaptation where the domain is a new modality (Chang et al., 2020; Kamboj & Do, 2024).

Currently, two approaches can address UMA:

1. Student-Teacher (ST) methods: These leverage a teacher modality to distill knowledge to the student modality, a technique commonly used for semi-supervised learning (Thoker & Gall, 2019; Xue et al., 2022; Kong et al., 2019; Wang et al., 2020; Bruce et al., 2021).
2. Contrastive Alignment (CA): This approach aligns latent representations of data samples across modalities and employs a shared task head for cross-modal transfer (Moon et al., 2022; Girdhar et al., 2023; Gong et al., 2023).

Further motivation for the term UMA, and a review of existing ST and CA methods for RGB to IMU transfer is provided in the Related Works (Section 6.1).

Although these methods work well for image and text modalities, when applied to sensor signals, they compress a full time-series sequence into a single latent representation for alignment. This approach does not account for the continuous nature of real-world data and the variable time spans over which actions occur. The proposed method, Cross-modal Transfer Through Time (C3T), (1) extracts a time-varying latent dimension through the feature maps outputted from temporal convolutions, (2) performs contrastive alignment across modalities in this temporal latent space, and (3) uses a shared self-attention head to perform the transfer. **C3T aligns local temporal features across modalities at a fine-grained level, avoiding the loss of complex temporal dependencies during transfer.**

Our research evaluates these three UMA methods—ST, CA, and C3T—focusing on knowledge transfer from RGB videos to IMU signals across four diverse HAR datasets. C3T outperforms existing methods by at least 8% in top-1 accuracy on all datasets. Additional experiments demonstrate

C3T's superior resilience to time-shift, misalignment, and time-dilation noise. This is particularly valuable for real-world continuous human activity recognition, where the start and the end of an activity are not well defined, and actions can occur at various speeds.

Furthermore, a qualitative analysis of the multi-modal representation space reveals that contrastive methods excel by uncovering latent correlations between modalities, enabling cross-modal information transfer without labeled data. Additional visualizations of C3T's attention weights demonstrate how temporal shifts are captured across the temporal latent vectors, suggesting its performance gain stems from its robustness to temporal dynamics.

The promising results of C3T open new avenues for developing highly generalizable models for time-series sensor data. This has implications across various domains, including healthcare monitoring, smart homes, industrial IoT, and human-computer interaction, where multi-modal learning from limited labeled sensor data is crucial (Kamboj & Do, 2024). **Our novel contributions are as follows:**

- A formulation of the Unsupervised Modality Adaptation (UMA) setting for Human Activity Recognition (HAR), with a categorization of existing methods into Student-Teacher (ST) and Contrastive Alignment (CA) approaches. We introduce Cross-modal Transfer Through Time (C3T), and a corresponding temporal alignment loss function $\mathcal{L}_{\text{C3T}}$ (Section 6.1 to Section 2).
- A comprehensive evaluation and comparison of ST, CA, and C3T methods in the UMA setting on clean data and with temporal noise. These experiments demonstrate C3T's superior performance and robustness (Section 3).
- An in-depth examination of cross-modal alignment and C3T's structure, including visualizations of the multimodal representation space, a comparison of model sizes, and additional experiments on training and testing methods. This examination reveals insights into effective cross-modal transfer and its potential real-world applications for sensor data (Section 3 and Section 4).

## 2 UNSUPERVISED MODALITY ADAPTATION METHODS

**Unsupervised Modality Adaptation:** In domain adaptation, a model trained in a source domain is tasked with efficiently adapting to a related target domain that contains fewer labeled data points (Pan & Yang, 2009; Farahani et al., 2021). In the context of IMU-based HAR, domain adaptation may involve adapting between different sensors, adjusting to varying positions of wearables on the body, accommodating different users, or adapting to different IMU device types (Kamboj & Do, 2024). For video domain adaptation, challenges include adapting between different datasets, handling varying lighting conditions, accommodating different camera viewpoints, and transferring across different action categories or video qualities (Xu et al., 2024). Our work focuses on unsupervised domain adaptation where the target domain is a new, completely unlabeled modality. Hence, we introduce the term Unsupervised Modality Adaptation (UMA). Other research works explore similar concepts, such as knowledge distillation, missing modality, robust sensor fusion, multimodal alignment; however, most of these works require *some* labels from the target modality in training to update the model, thus do not work in UMA (Garcia et al., 2018; Wang et al., 2020; Nugroho et al., 2023; Yang et al., 2022). We use the term UMA to discuss performing test-time

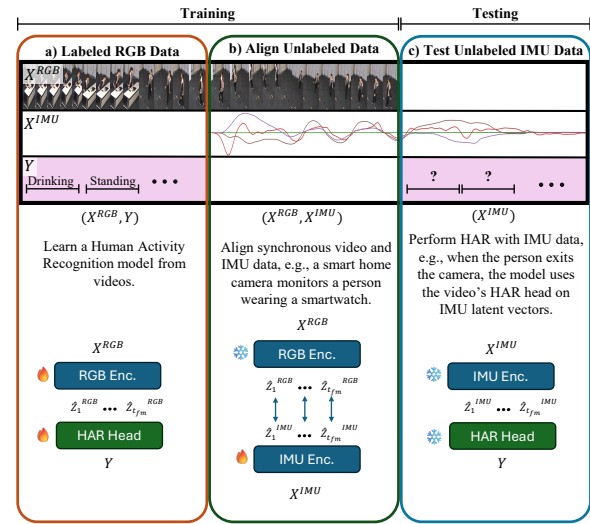

Figure 2: **Motivation for Unsupervised Modality Adaptation:** Depicts a scenario in which an AI system can perform inference on a new modality without additional annotation effort, through alignment with a pretrained modality. C3T leverages a temporal latent space for more robust transfer.

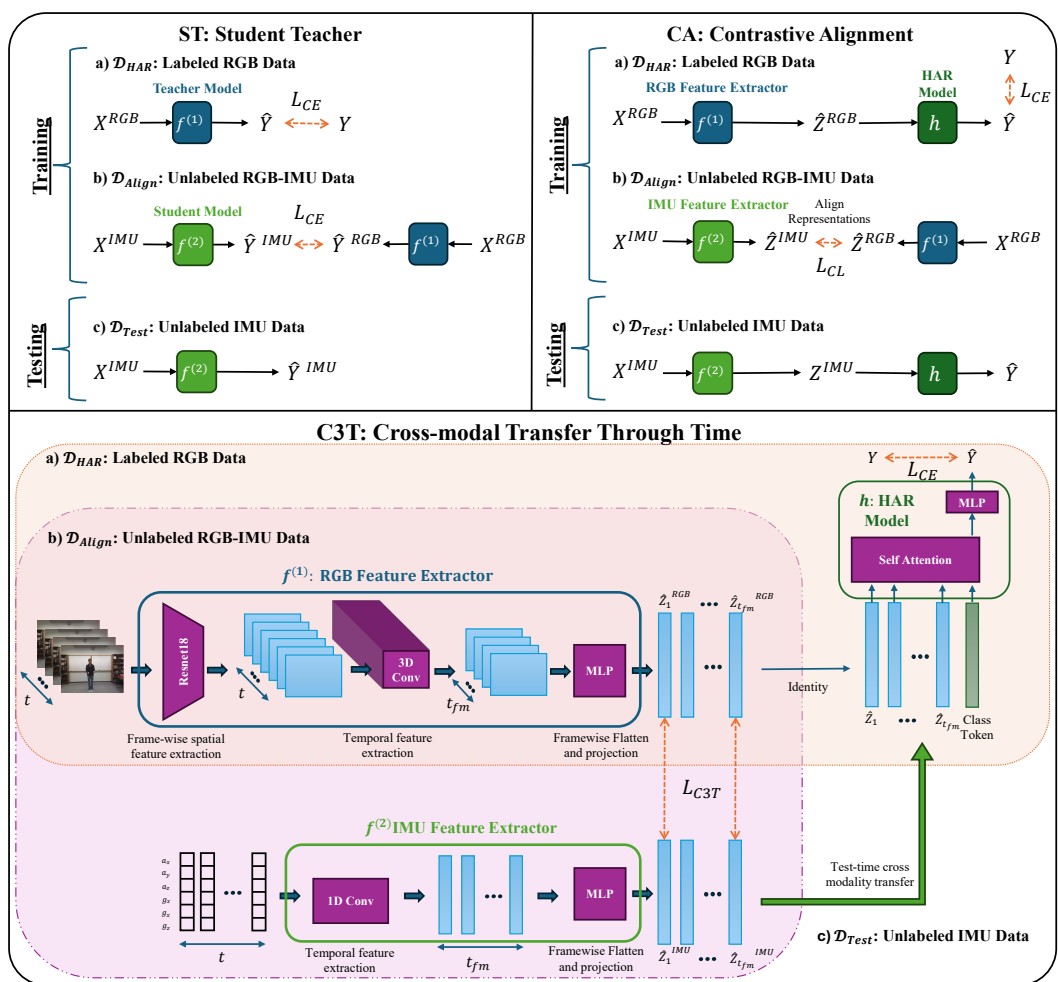

Figure 3: **Unsupervised Modality Adaptation Methods:** Training happens in two phases: a) trains the HAR model on labeled RGB inputs and b) aligns unlabeled IMU and RGB modalities. UMA testing c) occurs on unlabeled IMU data.

inference when *zero* labeled instances of the target modality are available during training. One such scenario is depicted in Figure 2.

In this section, we formulate and compare the 2 existing methods to perform UMA, Student-teacher and Contrastive Alignment. We then introduce our Cross-modal Transfer Through Time approach. In the context of human action or activity recognition (HAR), we conduct experiments using RGB videos as the source domain, $x^{(\text{RGB})}$, and Inertial Measurement Unit (IMU) data as the target $x^{(\text{IMU})}$. As depicted for each method in Figure 3, training for UMA occurs in two phases: a) Supervised Learning with RGB data on data split $\mathcal{D}_{\text{HAR}}$ and b) Unsupervised alignment across both modalities on $\mathcal{D}_{\text{Align}}$. Inference (phase c) occurs on IMU data on $\mathcal{D}_{\text{Test}}$.

**Student-Teacher:** We adopt a student-teacher (ST) method similar to Thoker & Gall (2019) for UMA. ST leverages the teacher RGB video model trained in *phase a* to produce pseudo-labels to train the student IMU model in *phase b* of Figure 3. In this case, the latent transfer space is the output logit space, $\mathcal{Z} = \mathcal{Y}$, or equivalently the HAR module is the identity $h = \mathbb{1}$, and the cross-modal representations are aligned using the standard cross-entropy loss $\mathcal{L}_{\text{CE}}$ (Appendix Equation (3)). We denote the teacher network as $f^{(1)} : \mathcal{X}^{(1)} \to \mathcal{Y}$ and the student network as $f^{(2)} : \mathcal{X}^{(2)} \to \mathcal{Y}$. First, we train the teacher $f^{(1)}$ on $\mathcal{D}_{\text{HAR}}$ with labeled RGB data. Next, since $\mathcal{D}_{\text{Align}}$ does not contain labels, we use $f^{(1)}(x_i^{(1)}) = \hat{y}_i^{(1)}$ to generate pseudo-labels for every datapoint $i$ in $\mathcal{D}_{\text{Align}}$. Then we use the augmented dataset $\hat{\mathcal{D}}_{\text{Align}} = \{(\mathbf{x}_i^{(1)}, \mathbf{x}_i^{(2)}, \hat{\mathbf{y}}_i^{(1)})\}_{i=1}^{I_2}$ to train $f^{(2)}$. The teacher network minimizes

$\mathcal{L}_{\mathrm{CE}}(P_{f^1(x)}, P_y)$ and the student minimizes $\mathcal{L}_{\mathrm{CE}}(P_{f^2(x)}, P_{\hat{y}_i^{(1)}})$, where $P$ represents the distribution generated by the labels, pseudo-labels or an encoder's output logits.

**Contrastive Alignment:** We adopt the contrastive alignment (CA) method from past works (Moon et al., 2022; Girdhar et al., 2023; Gong et al., 2023) for our UMA setting. In our work, CA performs *phase a* similar to ST; however, it trains a model with two parts: An encoder $f^{(1)}$ to extract the latent variable $z$, and a task-specific MLP head $h$. The extracted latent space $\mathcal{Z}$ enables scalability and interoperability for adding different sensing modalities, types of encoders, and output task heads.

In *phase b* of training, CA performs unsupervised contrastive alignment with the outputs of the RGB encoder $f^{(1)}$ and the IMU encoder $f^{(2)}$ on unlabeled data. To align different modalities in the feature space on $\mathcal{D}_{\mathrm{Align}}$ we use a symmetric contrastive loss formulation $\mathcal{L}_{CL}$ (Radford et al., 2021; Moon et al., 2022; Girdhar et al., 2023; Gong et al., 2023) with temperature parameter $\tau$ and batch size $B$:

$$\mathcal{L}_{\mathrm{CL}} = -\frac{1}{B}\sum_{i=1}^{B}\log\frac{\exp(\langle \hat{z}_i^{(1)}, \hat{z}_i^{(2)}\rangle/\tau)}{\sum_{j=1}^{B}\exp\left(\langle \hat{z}_i^{(1)}, \hat{z}_j^{(2)}\rangle/\tau\right)}, \text{where } \hat{z}_i^{(k)} = \frac{f^{(k)}(x_i^{(k)})}{||f^{(k)}(x_i^{(k)})||}, k \in \{1, 2\}. \quad (1)$$

Equation (1) clusters representations in $\mathcal{Z}$ by cosine similarity, which brings about the desired property of the latent space that semantically similar vectors are proximal.

**Cross-modal Transfer Through Time:** CA and ST do not leverage latent temporal information as they collapse the entire time sequence into one latent vector. We thus propose a Cross-modal Transfer Through Time (C3T) model that leverages the temporal information of time-series sensors when aligning their representations. For a given time series input $x_{i,t}^{(k)}$ for sample i in a dataset, timestep $t = 1 \dots T$ of modality $k$, C3T leverages an encoder $f^{(k)} : \mathcal{X}_T^{(k)} \to \mathcal{Z}_{t_{\mathrm{fm}}}$ to extract a set of $t_{\mathrm{fm}}$ latent vectors, $z_t$. from the feature map of temporal convolutions. Thus, the number of latent vectors is the length of the temporal dimension after convolution given by $t_{\mathrm{fm}} = \left\lfloor \frac{T+2P-D(K-1)-1}{S} + 1 \right\rfloor$ where P, D, K, and S are the padding, dilation, kernel size, and stride, respectively.

Furthermore, during the alignment phase, C3T aligns each of these latent time vectors with the same time vector from the other modality. The new loss function extends the contrastive loss formulation to compare every time step $t$ for the same data sample to all the other data samples and their time steps in the batch. Thus for a batch of size $B$ with the data for every time step $t$, element $i$ and modality $k$ being indicated by $x_{i,t}^{(k)}$, $\mathcal{L}_{\mathrm{C3T}}$ can be calculated as follows:

$$\mathcal{L}_{\mathrm{C3T}} = -\frac{1}{B}\sum_{i=1}^{B}\sum_{t=1}^{t_{\mathrm{fm}}}\log\frac{\exp(\langle \hat{z}_{i,t}^{(1)}, \hat{z}_{i,t}^{(2)}\rangle/\tau)}{\sum_{j=1}^{B}\sum_{l=1}^{t_{\mathrm{fm}}}\exp\left(\langle \hat{z}_{i,t}^{(1)}, \hat{z}_{j,l}^{(2)}\rangle/\tau\right)}, \text{where } \hat{z}_{i,t}^{(k)} = \frac{f^{(k)}(x_{i,t}^{(k)})}{||f^{(k)}(x_{i,t}^{(k)})||}, k \in \{1, 2\} \quad (2)$$

C3T enables the use of a self-attention HAR module to extract global temporal features after the alignment of local temporal features across modalities, as shown in Figure 3. C3T's HAR self-attention head, $h$, utilizes a class token similar to ViT (Dosovitskiy et al., 2020), with temporal feature map vectors serving as input in place of image patches.

# 3 CROSS-MODAL TRANSFER EXPERIMENTS

**Implementation:** Our experiments utilize three key neural network modules:

1. *Video Feature Encoder* $f^{(1)}$: We employ a pretrained ResNet18 on every frame of the video followed by 3D convolutions and a 2-layer multi-layer perceptron (MLP) with ReLU activations. Resnet is a well-established lightweight spatial feature extractor (Hara et al., 2017), and 3D convolutions are effective temporal feature extractors (Tran et al., 2018; Carreira & Zisserman, 2017).
2. *IMU Feature Encoder* $f^{(2)}$: A 1D CNN followed by an MLP is used here. CNNs have shown superior performance in extracting features from time-series data like IMU signals, efficiently capturing local patterns and temporal dependencies (Valarezo et al., 2017).
3. *HAR Task Decoder* $h$: ST does not require $h$ and CA uses an MLP. C3T employs a self-attention module to better capture long-range dependencies in the latent space, which is particularly beneficial for complex action sequences (Moutik et al., 2023).

Table 1: **UMA vs. Supervised Performance:** The supervised baselines train, using labels for both modalities. The UMA methods train with labeled RGB data and separate unlabeled RGB+IMU data. The Top-1 and Top-3 accuracies on $\mathcal{D}_{\text{Test}}$ are shown. C3T outperforms the other UMA methods and nears the performance of the supervised setting, in four diverse datasets capturing various scenes, occlusions, IMU placements, and camera views. † (Thoker & Gall, 2019) ∗ (Gong et al., 2023)

| | Model | UTD-MHAD | | CZU-MHAD | | MMACT | | MMEA-CL | | Overall Average |
|---|---|---|---|---|---|---|---|---|---|---|
| | | Top-1 | Top-3 | Top-1 | Top-3 | Top-1 | Top-3 | Top-1 | Top-3 | |
| Supervised | IMU | **87.9** | **97.7** | **95.1** | 98.2 | 70.0 | 90.0 | 65.8 | 87.6 | **86.7** |
| | RGB | 53.8 | 73.1 | 94.0 | **99.7** | 42.1 | 61.6 | 54.2 | 77.1 | 68.9 |
| | Fusion | 62.5 | 82.2 | 95.0 | 98.5 | **76.7** | **92.0** | **80.1** | **92.7** | 85.2 |
| UMA | Random | 3.7 | 11.1 | 4.6 | 16.6 | 2.9 | 8.6 | 3.1 | 9.4 | 7.5 |
| | ST † | 12.9 | 24.6 | 41.1 | 61.9 | 17.6 | 34.7 | 9.9 | 22.7 | 27.9 |
| | CA ∗ | 42.6 | 67.4 | 70.0 | 92.7 | 24.5 | 47.6 | 29.3 | 51.7 | 52.0 |
| | **C3T (Ours)** | 62.5 | 86.4 | 84.2 | 96.7 | 32.4 | 57.9 | 51.2 | 78.8 | 65.0 |

These simple models with comparable module sizes across methods isolate the impact of C3T's novel technique, demonstrating its efficacy independent of complex architectures or larger datasets. This approach facilitates potential future scaling and expansion of the method.

**Datasets and Hyperparameters:** We present results on four diverse datasets: (1) UTD-MHAD (Chen et al., 2015), a small yet structured dataset; (2) CZU-MHAD (Chao et al., 2022), a slightly larger dataset captured in a controlled environment; (3) MMACT (Kong et al., 2019), a very large dataset with various challenges including different view-angles, scenes, and occlusions; and (4) MMEA-CL (Xu et al., 2023), an egocentric camera dataset. For each dataset, we create an approximately 40-40-10-10 percent data split for the $\mathcal{D}_{\text{Align}}$, $\mathcal{D}_{\text{HAR}}$, $\mathcal{D}_{\text{Val}}$, and $\mathcal{D}_{\text{Test}}$ sets respectively, as shown in Appendix Table 9. $\mathcal{D}_{\text{Val}}$ was used to perform a minor hyperparameter search on the UTD-MHAD dataset. The methods performed best with a learning rate of $1.5 \times 10^{-4}$, a batch size of 16, and a latent representation dimension of 2048 with an Adam optimizer. The preprocessing steps downsample the video to $t = 30$ frames, and C3T extracts $t_{\text{fm}} = 15$ latent vectors per sample. Experiments were implemented in Pytorch and run on a 16GB NVIDIA Quadro RTX 5000, four 48GB A40s, or four 48GB A100s. More detailed information about each dataset and implementation can be found in Appendix Section 6.8.

**UMA vs Supervised Settings:** Our results in Table 1 show Top-1 and Top-3 test accuracies on 4 different datasets. Along with the UMA setting, we show the results of our architecture under complete supervision: the *IMU* row trains $f^{(1)} : \mathcal{X}^{(\text{IMU})} \rightarrow \mathcal{Y}$, *RGB* trains $f^{(2)} : \mathcal{X}^{(\text{RGB})} \rightarrow \mathcal{Y}$, and *Fusion* averages the outputs of $f^{(1)}$ and $f^{(2)}$ and trains a linear head $h$. The *Random* row shows the probability of guessing the correct class, assuming a uniform distribution. We run each experiment thrice with different random seeds, reporting the average accuracies to ensure rigorous empirical results. Appendix Figure 12 shows all the runs, the computed averages, and their corresponding standard deviations.

The experimental results in the UMA setting consistently rank C3T as the most accurate, followed by CA, and then ST. Despite its lower ranking, ST shows a 3-10x improvement over random prediction, indicating some efficacy in UMA. We attribute ST's limited performance to its reliance on the teacher model's accuracy and its inability to leverage multimodal correlations in a latent space. Figure 6 supports this conclusion by illustrating how contrastive alignment encourages multimodal representations to cluster by class prior to label introduction. We hypothesize that C3T outperforms CA by aligning modalities across a larger temporal representation space, enabling it to capture more detailed temporal information. Finally, another notable result from Table 1 is that on UTD-MHAD Top-1 and MMEA-CL Top-3 accuracies, C3T surpasses the supervised RGB model. This may imply that cross-modal temporal alignment may uncover an inherent correlation between the modalities more informative than the label information.

**Accuracy vs Latent Size:** Figure 4 illustrates the UMA performance across different latent vector sizes. Since ST does not utilize a latent vector, we adjust the hidden size of its final MLP, observing relatively consistent performance. Furthermore, CA's performance declines significantly with smaller latent vectors, whereas C3T maintains superior performance even at reduced model sizes. Notably, C3T's size scales efficiently as the self-attention head size decreases significantly with reduced input

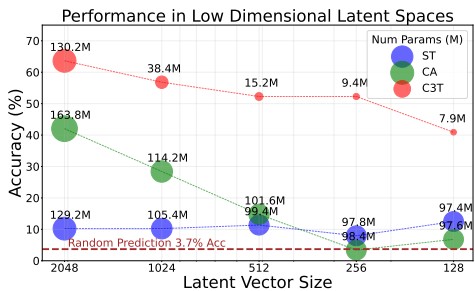

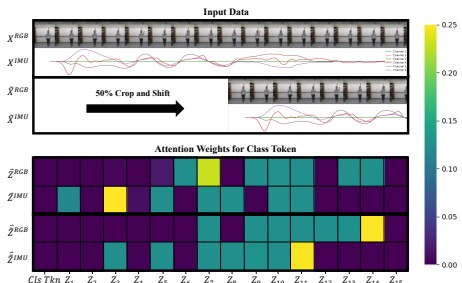

Figure 4: **Accuracy vs Latent Size:** Accuracy decreases with latent vector size. C3T maintains superior performance in scaling to smaller models.

Figure 5: **Attention heatmap for C3T's HAR module:** Input shifts shift the attention weights of the temporal latent vectors.

Table 2: **Noise Experiments**: Performance of ST, CA, and C3T under various noise conditions. All results report UMA accuracy (↑) and its percent difference (↓) from the Original (no noise) column. † (Thoker & Gall, 2019) ∗ (Gong et al., 2023)

| Model | Original | 1. Crop | 2. Misalign | 3. Dilate | 4. All |
|---|---|---|---|---|---|
| ST † | 12.9 | 3.4 (-73.6%) | 5.7 (-55.8%) | 5.7 (-55.8%) | 10.2 (-20.9%) |
| CA ∗ | 42.6 | 10.2 (-76.1%) | 2.3 (-94.6%) | 21.6 (-49.3%) | 18.2 (-57.3%) |
| C3T | **62.5** | **52.3** (-16.3%) | **46.6** (-25.4%) | **56.8** (-9.1%) | **58.0** (-7.2%) |

token dimensions, underscoring C3T's suitability for resource-constrained environments such as on-device computing for wearables or smart devices.

**UMA in Temporal Noise:** We evaluated each method's robustness to temporal noise during testing, simulating three real-life scenarios (Table 2):

1. **Crop:** Randomly shifts and crops both modalities' time sequences by up to 60%, simulating continuous real-time action recognition, where an action may not occur in the middle of the time sequence.
2. **Misalign:** Applies crop to one modality, mimicking hardware asynchrony or differing frame rates.
3. **Dilation:** Applies crop to both modalities and then upsamples, imitating slower action movements.
4. **All:** Applies all the types of noise to each data sample.

C3T demonstrates robust performance under temporal noise, likely due to its attention-based HAR module leveraging tokens generated by the feature map of a temporal convolution. The self-attention mechanism compares neighboring tokens from various time sections, effectively capturing actions regardless of their temporal position within the sequence. Figure Figure 5 visualizes the attention weights of the latent vectors in C3T's self-attention head when the input sample is cropped and shifted. Interestingly, **the shifted input results in a corresponding shift in the attention scores, illustrating its robustness toward this noise pattern.** Additionally, due to the design of the self-attention block, C3T's HAR head is invariant to variable-length inputs during inference, which provides an advantage in cropped scenarios, requiring minimal zero-padding compared to the ST and CA methods.

**Ablations:** To demonstrate that C3T's performance advantage stems from its temporal alignment approach rather than its self-attention HAR module, we conducted comprehensive ablation studies on UTD-MHAD. Our results show two key findings: (1) adding attention mechanisms to ST or CA frameworks fails to match C3T's performance, and (2) C3T maintains superior performance over both methods even without its attention module.

First, we compare UMA performance using convolution and attention-based feature extractors. The RGB encoder has two feature extraction steps (spatial and temporal) and the IMU has one (temporal). Table 3 indicates that ST or CA with attention do not perform as well even with larger parameter sizes. This aligns with previous work showing that convolutions are superior for IMU data (Valarezo et al., 2017), and visual attention works less well on small-scale datasets (Dosovitskiy et al., 2020).

We further ablate C3T head architectures (Table 4). We test two attention-based heads and two MLP heads that condense the $Z_1 \ldots Z_{t_{rec}}$ latent vectors into a single output class. The first attention head uses a class token as shown in Figure 3, and the second concatenates all the output tokens and projects it to an output. The MLP methods either add or concatenate all the temporal features before passing it through an MLP. While concatenating latent vectors and using an MLP performed best on clean data

Table 3: **Architecture Ablation:** Encoder types are reported as (RGB-Spatial / RGB-Temporal / IMU-Temporal), where C = Convolution and A = Attention.

| Method | Encoders | Params (M) | Acc. (%) |
|--------|----------|-----------|----------|
| ST | C/C/C | 129.2 | **12.9** |
|    | C/C/A | 97.8 | 10.2 |
|    | C/A/C | 871.2 | 11.4 |
|    | A/C/C | 291.5 | 5.7 |
| CA | C/C/C | 163.8 | **42.6** |
|    | C/C/A | 132.3 | 19.3 |
|    | C/A/C | 905.7 | 34.1 |
|    | A/C/C | 326.0 | 26.1 |
| C3T | C/C/C | 137.7 | **62.5** |
|     | C/C/A | 106.3 | 15.9 |
|     | C/A/C | 879.6 | 53.4 |
|     | A/C/C | 300.0 | 33.0 |

Table 4: **C3T HAR Module Ablations:** Comparison of 2 Attention methods and two MLP methods.

| Input | Attention | | MLP | |
|-------|-----------|---------|-----|---------|
|       | Cls Token | Concat. | Add | Concat. |
| Clean | 62.5 | 44.3 | 56.8 | **70.5** |
| Noisy | **52.3** | 43.2 | 50.0 | 43.2 |

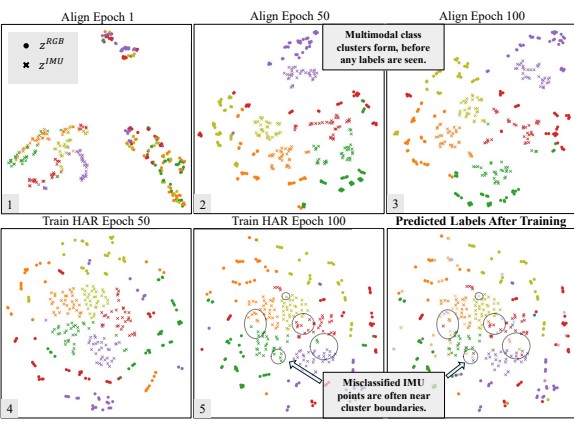

Figure 6: **CA TSNE Plots in UMA:** The following shows the progression of the latent representations of datapoints for 5 classes (Bowling, Clap, Draw circle (clockwise), Jog, Basketball shoot) during training CA on the UTD-MHAD dataset. In the end, we plot the predicted labels and circle areas of confusion, which seem to often occur at the boundaries between clusters.

(70.45%), the class token mechanism offered superior robustness to noise and was thus chosen for C3T. **Furthermore, all C3T variants in Table 4 outperform CA and ST in UMA performance (12.9% and 42.6% respectively from Table 1), emphasizing C3T's strength lies in temporal alignment, not its attention mechanism.** The key distinction between CA and C3T+Concat-MLP is where the feature vectors are combined: CA condenses the convolution output map to one feature vector before alignment, while C3T does so post-alignment for the HAR task. This suggests that effective cross-modal knowledge transfer at the local feature level is crucial for time-series data, with global features being more appropriately utilized for post-transfer inference.

## 4 ADDITIONAL EXPERIMENTS AND DISCUSSION

***How do we train the CA and C3T Architectures?***

(1) **Align First:** First aligns the representations generated by the RGB and IMU encoders, $f^{(1)}$ and $f^{(2)}$, on $\mathcal{D}_{\text{Align}}$ (*phase b* depicted in Figure 3). Then it freezes the weights for both encoders and trains the HAR module $h$ on RGB data in $\mathcal{D}_{\text{HAR}}$ (*phase a* in Figure 3).

(2) **HAR First:** Reverses the order from 1. First it performs *phase a* (RGB HAR training), then it freezes the RGB encoder $f^{(1)}$ and aligns the IMU encoder $f^{(2)}$ (*phase b*).

(3) **Interspersed Training:** Intermittently learns from $\mathcal{D}_{\text{Align}}$ and $\mathcal{D}_{\text{HAR}}$. The model learns an epoch from *phase a* and updates its weights to train the RGB HAR model, then learns an epoch from *phase b* and updates its weights to align the encoders, iterating between the two losses.

(4) **Combined Loss:** Train both *phase a* and *phase b* but within the same loss iteration. The loss from *phase a* on a batch of data from $\mathcal{D}_{\text{HAR}}$ is added to the loss from *phase b* on a batch of data from $\mathcal{D}_{\text{Align}}$ and the total is used to update the weights: $\mathcal{L}_{\text{Total}} = \mathcal{L}_{\text{CE}} + \mathcal{L}_{\text{CL/C3T}}$.

**Results:** As shown in the Table 5, training by the *Align First* method performs the best for C3T, whereas *Combined Loss* performs the best for CA. The main experiments reported in this work (Table 1) use training methods *Align First* and *Combined Loss* for C3T and CA, respectively. We hypothesize that *HAR First* yields a latent space tailored to RGB HAR, which does not capture IMU-RGB correlations, and *Interspersed Training* faces instability in training.

Figure 6 visualizes the latent space outputs of CA training with the *Align First* method using t-SNE plots (Van der Maaten & Hinton, 2008). The model quickly segments classes during the align phase,

Table 5: **Additional Experiments**: Performance of ST, CA, and C3T across various training methods, testing modalities, and reverse transfer from IMU to RGB data (RT). See Section 4 for details.

| Model | Training Method | | | | Modality Testing | | | RT |
|---|---|---|---|---|---|---|---|---|
| | (1) | (2) | (3) | (4) | (1) IMU | (2) RGB | (3) Both | I→R |
| ST | - | - | - | - | 12.9 | 53.8 | 17.0 | 4.60 |
| CA | 38.6 | 36.4 | 27.3 | **42.6** | 42.6 | 56.8 | 60.2 | 27.3 |
| C3T | **62.5** | 35.2 | 51.1 | 27.9 | **62.5** | **78.4** | **79.5** | **31.8** |

even without labels, suggesting that the data's natural structure facilitates class distinction across different modalities. This supports the Platonic Representation Hypothesis (Huh et al., 2024) which posits that the same semantic concepts learned from different modalities or data-views are converging to some ground-truth concept space. The separable clusters also imply that CA and C3T could potentially adapt to new class labels with just a few samples, as the latent structure would have already grouped similar classes. Furthermore, after training, the model tends to misclassify points near the boundary between clusters, implying that the HAR head is learning a decision boundary in this latent space. Figure 6 supports our initial hypothesis (Figure 2) that a joint latent space could be leveraged to perform UMA using a classification head trained only on RGB data.

***Can UMA methods retain performance on the labeled modality or leverage both modalities?***

(1) **IMU (UMA):** Is the main result of this paper and described above in Section 2.
(2) **RGB (Supervised Learning):** Tests the model on RGB data, which was labeled during training.
(3) **Both (Sensor Fusion):** Merges latent vectors from each modality by adding them.

**Results:** Table 5 shows that C3T outperforms the other methods. When comparing the performances in the different test scenarios, our experiments indicate that when given both modalities, fusion performs better than the RGB model alone. Instead of introducing noise or uncertainty into the model, introducing an unlabeled modality may add structure to the shared latent space that bolsters performance, especially if that modality is highly informative for the given task. This observation bears resemblance to knowledge distillation methods, where an auxiliary modality during training leads to improved testing outcomes; however, these methods usually assume that auxiliary modality is labeled (Chen et al., 2023). Applying UMA to sensor fusion is a promising future direction.

***Can the transfer be performed in reverse?***

To transfer from IMU to RGB data, C3T trains on labeled IMU data and unlabeled IMU+RGB data, then tests on unlabeled RGB data. The results in the I→R column of Table 5 indicate that C3T's superior performance generalizes to this new configuration. Furthermore, the performance is worse when transferring from IMU to RGB, implying that transfer from a more informative data modality (RGB) to a less informative one (IMU) is more effective. This aligns with our initial motivation that RGB data is more abundant and easier to label, and RGB to IMU has more real-world applications, such as transferring from a smart home camera to a smartwatch as outlined in Figure 2. Nonetheless, further analysis of C3T in various scenarios and modalities is a relevant future research extension.

The supplementary Appendix contains current status of existing related literature 6.1, a deeper discussion of future directions and limitations (Section 6.4) as well as additional visualizations (Section 6.3), experiments (Section 6.5), ablations (Section 6.6), implementation details (Sections 6.7 and 6.8) and baselines (Section 6.9).

# 5 CONCLUSION

This work explores Unsupervised Modality Adaptation (UMA) for human activity recognition, challenging models to perform inference on a modality that was unlabeled during training. Our experiments focused on constructing a unified latent space between modalities and comparing three UMA methods in various settings. Our Cross-modal Transfer Through Time (C3T) method performs alignment on a more fine-grained level and shows significant improvements and robustness for RGB to IMU cross-modal transfer. We hope that our results inspire further exploration of temporal latent spaces for robust sensor-based cross-modal transfer.

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

# 6 APPENDIX: RELATED WORKS, ADDITIONAL DESCRIPTION AND VISUALIZATIONS

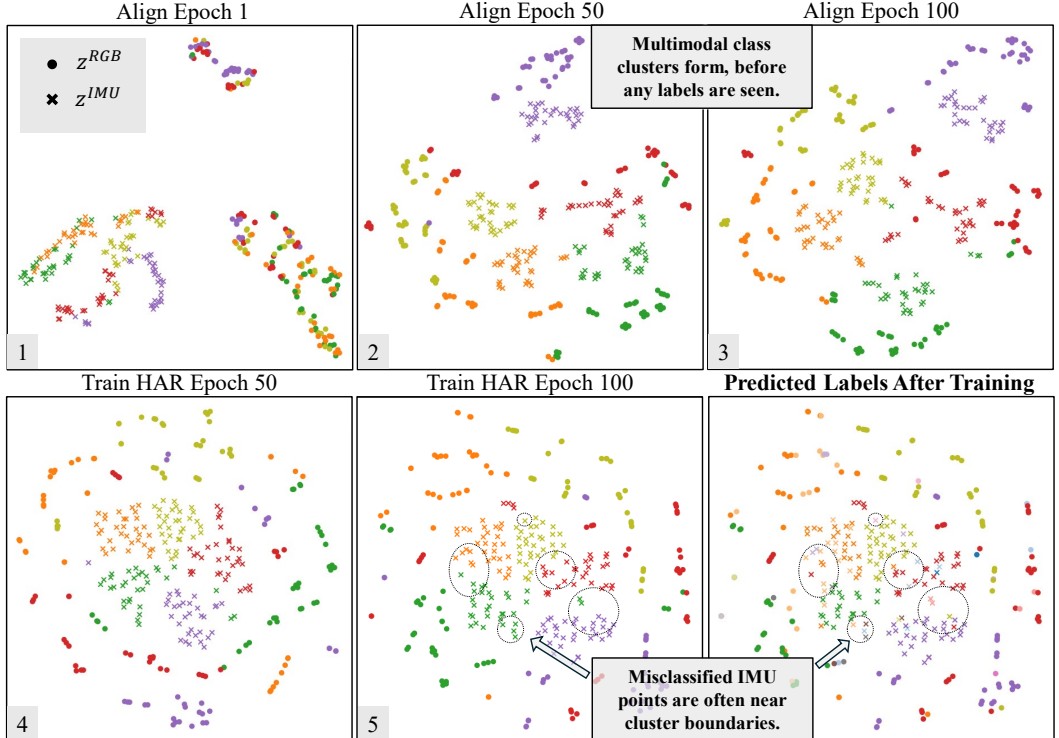

Figure 7: **CA TSNE Plots in UMA:** (Reproduced Figure 6 Larger) The following shows the progression of the latent representations of datapoints for 5 classes (Bowling, Clap, Draw circle (clockwise), Jog, Basketball shoot) during training CA on the UTD-MHAD dataset. In the end, we plot the predicted labels and circle areas of confusion, which seem to often occur at the boundaries between clusters.

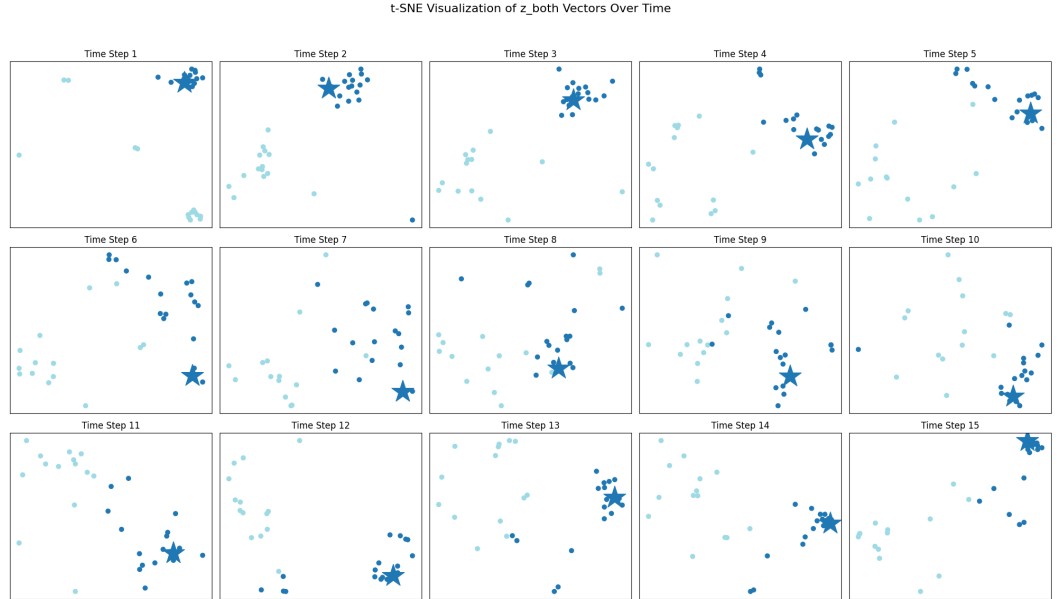

Figure 8: **C3T TSNE plot with shifted input:** This plot visualizes the TSNE plots of the $t_{rec} = 15$ latent $z_i = 0 \dots t_{rec}$ for various points of two classes. The $z$s shown are the fused representation between the IMU and RGB modalities. The dark blue star is a point that was *shifted by 50%* compared to the rest of the visualized points for the class. Notice how some time steps are more distinctive than others between the classes, and the star tends towards the edge of the group for many $z$s.

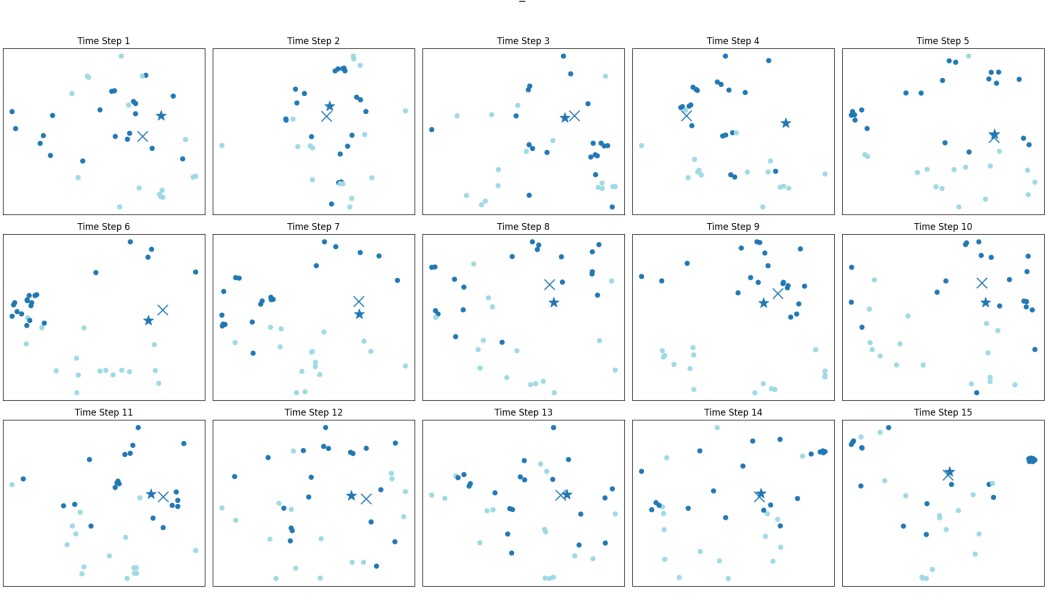

Figure 9: **C3T TSNE plot with shifted input:** Another TSNE plot with shifted input for different classes from the previous figure (Dark blue shows 'Swipe left' and light blue shows 'Pick up and throw'). This time X marks the regular input and star marks the shifted inputs. Unexpectedly, they are relatively close through out, indicating the temporal convolutions that constructed these z's might be doing part of the work when accounting for robustness to shift noise. Furthermore, notice how $z_7$to$z_{10}$ indicate latent variables that are more distinctive (there is a more clear boundary between the light blue and dark blue points). This could indicate that these time steps are most important for distinguishing between the two given classes.

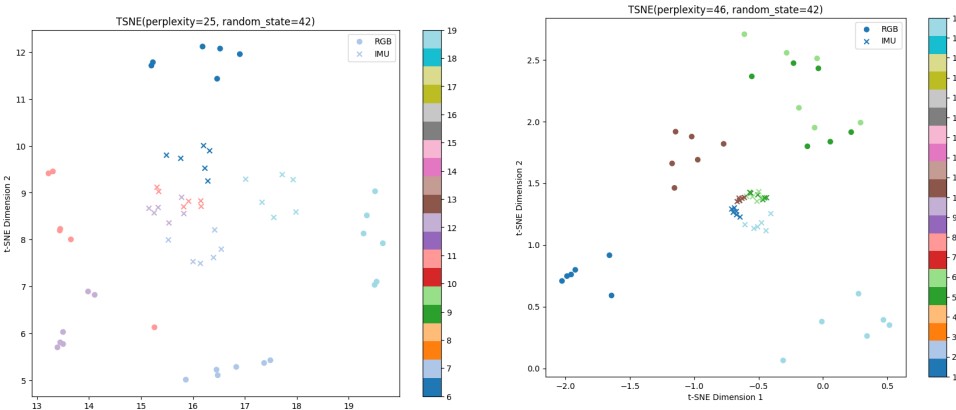

Figure 10: **CA TSNE on CZU Dataset:** These plots indicate the same trend discussed in the main paper, that the IMU points in the multimodal representation space tend to cluster in the middle and mirror the RGB points on the outside. Particularly, for the CZU dataset the IMU signals are stronger (60 IMU channels, 6 on 10 wearable devices) and this clustering tends to be stronger.

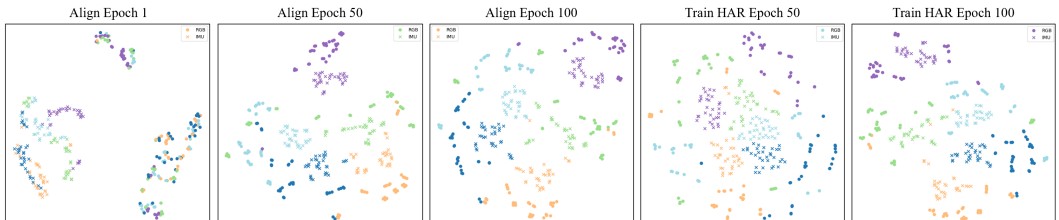

Figure 11: **CA TSNE Plots:** These are similar to the plot shown in the main paper Figure 6, but with different classes. The following shows the progression of the latent representations for 5 classes (5 different colors) during training CA on the UTD-MHAD dataset in UMA. Circles indicate RGB data, and crosses indicate IMU data points.

Figure 12 (rotated landscape table). Full results across all trials for Table 1.

| | | UTD Top1 | Top3 | Top5 | Top7 | MMACT Top1 | Top3 | Top5 | Top7 | MMEA Top1 | Top3 | Top5 | Top7 | CZU Top1 | Top3 | Top5 | Top7 |
|---|---|---|---|---|---|---|---|---|---|---|---|---|---|---|---|---|---|
| **Supervised RGB** | trial 1 | 61.36364 | 75 | 79.54546 | 82.95454 | 42.37089 | 63.26291 | 74.53052 | 82.39436 | 56.70732 | 79.87805 | 86.28049 | 91.46342 | 94.64286 | 100 | 100 | 100 |
| | trial 2 | 61.36364 | 79.54546 | 86.36364 | 90.90909 | 41.19718 | 59.50704 | 69.71831 | 79.10798 | 56.09756 | 78.5061 | 85.97561 | 89.93903 | 96.42857 | 100 | 100 | 100 |
| | trial 3 | 38.63636 | 64.77273 | 67.04546 | 72.72727 | 42.60563 | 62.0892 | 72.1831 | 78.05164 | 49.84756 | 72.86585 | 80.18293 | 85.97561 | 91.07143 | 99.10714 | 99.10714 | 99.70238 |
| | avg | 53.78788 | 73.10606 | 77.65152 | 82.19697 | 42.0579 | 61.61972 | 72.14398 | 79.85133 | 54.21748 | 77.08333 | 84.14634 | 89.12602 | 94.04762 | 99.70238 | 99.70238 | 100 |
| | std | 13.1216 | 7.566283 | 9.797361 | 9.114553 | 0.754589 | 1.921445 | 2.406344 | 2.26478 | 3.796723 | 3.716307 | 3.4358 | 2.8328 | 2.727723 | 0.515493 | 0.515493 | 0 |
| **IMU** | trial 1 | 87.5 | 97.72727 | 97.72727 | 97.72727 | 69.24883 | 90.14085 | 95.30516 | 96.83099 | 65.70122 | 87.5 | 92.68293 | 96.19903 | 94.64286 | 98.21429 | 98.21429 | 100 |
| | trial 2 | 87.5 | 97.72727 | 97.72727 | 97.72727 | 70.30516 | 90.14085 | 94.95305 | 96.71362 | 66.46342 | 87.80488 | 93.14024 | 95.27439 | 95.53571 | 98.21429 | 98.21429 | 100 |
| | trial 3 | 88.63636 | 97.72727 | 97.72727 | 97.72727 | 70.30516 | 89.78873 | 94.95305 | 96.47887 | 65.2439 | 87.5 | 92.53049 | 95.42683 | 94.64286 | 98.21429 | 100 | 100 |
| | avg | 87.87879 | 97.72727 | 97.72727 | 97.72727 | 69.95305 | 90.02348 | 95.07042 | 96.67449 | 65.80285 | 87.60163 | 92.78455 | 95.63008 | 94.94048 | 98.21429 | 98.80953 | 100 |
| | std | 0.656078 | 1.74E-14 | 1.74E-14 | 1.74E-14 | 0.608872 | 0.203297 | 0.203291 | 0.179291 | 0.616079 | 0.176023 | 0.317324 | 0.490026 | 0.515487 | 0 | 1.03098 | 0 |
| **FUSION** | trial 1 | 64.77273 | 87.5 | 92.04546 | 94.31818 | 77.34742 | 92.13615 | 94.71831 | 96.12676 | 83.84146 | 94.96951 | 96.95122 | 97.40854 | 95.53571 | 98.21429 | 99.10714 | 100 |
| | trial 2 | 51.13636 | 76.13636 | 81.81818 | 86.36364 | 75.93896 | 90.96244 | 95.89202 | 97.65258 | 71.18903 | 88.26219 | 92.22561 | 94.05488 | 93.75 | 99.10714 | 100 | 100 |
| | trial 3 | 71.59091 | 82.95454 | 88.63636 | 89.77273 | 76.87794 | 92.95775 | 95.53991 | 96.3615 | 85.21342 | 94.81707 | 96.79878 | 97.40854 | 95.53571 | 98.21429 | 99.10714 | 99.10714 |
| | avg | 62.5 | 82.19697 | 87.5 | 90.15152 | 76.72144 | 92.01878 | 95.38341 | 96.71361 | 80.0813 | 92.68292 | 95.3252 | 96.29065 | 94.94047 | 98.51191 | 99.40476 | 99.70238 |
| | std | 10.41495 | 5.719573 | 5.207475 | 3.990775 | 0.717153 | 1.00282 | 0.602302 | 0.821596 | 7.731427 | 3.829226 | 2.685408 | 1.936237 | 1.03098 | 0.515487 | 0.515493 | 0.515493 |
| **UMA ST** | trial 1 | 15.90909 | 27.27273 | 32.95454 | 37.5 | 17.60563 | 37.0892 | 48.94366 | 56.69014 | 9.60366 | 23.17073 | 30.79268 | 38.71951 | 40.17857 | 59.82143 | 70.53571 | 79.46429 |
| | trial 2 | 13.63636 | 23.86364 | 30.68182 | 30.68182 | 18.5446 | 36.2676 | 47.30047 | 55.86855 | 9.29878 | 20.73171 | 27.7439 | 37.19512 | 41.07143 | 64.28571 | 69.64286 | 74.10714 |
| | trial 3 | 9.09091 | 22.72727 | 30.68182 | 35.22727 | 16.66667 | 30.86855 | 38.7324 | 46.94836 | 10.82317 | 24.2378 | 34.7661 | 43.44512 | 41.96429 | 61.60714 | 68.75 | 77.67857 |
| | avg | 12.87879 | 24.62121 | 31.43939 | 34.4697 | 17.60563 | 34.74178 | 44.99218 | 53.16902 | 9.908537 | 22.71341 | 31.09756 | 39.78658 | 41.07143 | 61.90476 | 69.64286 | 77.08333 |
| | std | 3.471647 | 2.365532 | 1.312156 | 3.471647 | 0.938965 | 3.37938 | 5.48303 | 5.402886 | 0.806631 | 1.797226 | 3.516028 | 3.258774 | 0.892286 | 2.246972 | 0.892855 | 2.727727 |
| **CA** | trial 1 | 40.69767 | 66.27907 | 79.06977 | 84.88372 | 23.7927 | 47.34982 | 59.71732 | 73.38045 | 29.09648 | 50.38285 | 61.8683 | 69.67841 | 66.36364 | 94.54546 | 98.18182 | 100 |
| | trial 2 | 38.37209 | 60.46511 | 70.93023 | 80.23256 | 27.79741 | 52.06125 | 65.01767 | 75.02945 | 30.62787 | 55.43645 | 65.3905 | 71.97549 | 79.09091 | 94.54546 | 100 | 100 |
| | trial 3 | 48.83721 | 75.5814 | 86.04651 | 93.02225 | 21.90813 | 43.4629 | 59.95288 | 68.55124 | 28.17764 | 49.31087 | 62.17458 | 71.05666 | 64.54546 | 89.09091 | 94.54546 | 96.36364 |
| | avg | 42.63566 | 67.44186 | 78.68217 | 86.04651 | 24.49941 | 47.62466 | 61.56262 | 72.32038 | 29.30066 | 51.71006 | 63.14446 | 70.90352 | 70 | 92.72728 | 97.57576 | 98.78788 |
| | std | 5.495137 | 7.624934 | 7.56559 | 6.474141 | 3.007572 | 4.305759 | 2.994475 | 3.366692 | 1.237811 | 3.271359 | 1.951147 | 1.156172 | 7.925269 | 3.149186 | 2.777316 | 2.099453 |
| **C3T** | trial 1 | 59.09091 | 88.63636 | 92.04546 | 96.59091 | 33.21596 | 59.38967 | 70.77465 | 77.93427 | 49.08537 | 78.04878 | 83.84146 | 88.41463 | 81.25 | 97.32143 | 99.10714 | 99.10714 |
| | trial 2 | 60.22727 | 84.09091 | 92.04546 | 94.31818 | 32.277 | 58.4507 | 70.53991 | 77.58216 | 51.21951 | 78.96342 | 84.60366 | 88.87195 | 90.17857 | 96.42857 | 99.10714 | 99.10714 |
| | trial 3 | 68.18182 | 86.36364 | 90.90909 | 95.45454 | 31.69014 | 55.98592 | 66.90141 | 72.65258 | 53.35366 | 79.2683 | 85.36585 | 88.7195 | 81.25 | 96.42857 | 98.21429 | 100 |
| | avg | 62.5 | 86.36364 | 93.18182 | 95.45454 | 32.39437 | 57.9421 | 69.40532 | 76.05634 | 51.21951 | 78.76017 | 84.60366 | 88.7195 | 84.22619 | 96.72619 | 98.80952 | 99.40476 |
| | std | 4.953296 | 2.272725 | 3.006535 | 1.136365 | 0.769651 | 1.75795 | 2.171627 | 2.952993 | 2.134145 | 0.634658 | 0.762195 | 0.264034 | 5.154912 | 0.515493 | 0.515487 | 0.515493 |

Figure 12: **Full results:** across all trials for Table 1. Values that we used in Table 1 are highlighted and the best performance of each method within supervised and UMA are bolded.

In this appendix, we first discuss related studies and directions 6.1, then provide additional additional visualizations given above and summarize them in Section 6.3. Then, we discuss the limitations and future direction of this work. Further, we provide additional experiments in Section 6.5, additional ablations in Section 6.6, and further details on our methods in Section 6.7. Finally, we provide details on each of the datasets used in Section 6.8 and explore additional baselines to compare our methods to in Section 6.9.

## 6.1 RELATED WORKS

**Unsupervised Modality Adaptation** In domain adaptation, a model trained in a source domain is tasked with efficiently adapting to a related target domain that contains fewer labeled data points (Pan & Yang, 2009; Farahani et al., 2021). Given the focus on domains with scarce labels, adaptation is often achieved through unsupervised (Chang et al., 2020) or semi-supervised (An et al., 2021) methods. In the context of IMU-based HAR (Kamboj & Do, 2024), domain adaptation may involve adapting between different sensors (Bhalla et al., 2021), adjusting to varying positions of wearables on the body (Wang et al., 2018; Chang et al., 2020; Prabono et al., 2021), accommodating different users (Hu et al., 2023; Fu et al., 2021), or adapting to different IMU device types (Khan et al., 2018; Zhou et al., 2020; Chakma et al., 2021). Our work focuses on unsupervised domain adaptation where the target domain is a new, completely unlabeled modality. Hence, we introduce the term Unsupervised Modality Adaptation (UMA). Other research works explore similar concepts, such as knowledge distillation, missing modality, robust sensor fusion, multimodal alignment; however, most of these works require *some* labels from the target modality in training to update the model, thus do not work in UMA (Garcia et al., 2018; Wang et al., 2020; Nugroho et al., 2023; Yang et al., 2022). We use the term UMA to discuss performing test-time inference when *zero* labeled instances of the target modality are available during training.

**Student-Teacher:** Student-teacher methods involve two distinct models: a teacher (source) model that imparts knowledge to a student (target) model during the training process. Knowledge distillation methods often employ an extra auxiliary modality as the teacher during training to increase the student modality's performance during testing; however, they assume the availability of labeled training data from both modalities (Xue et al., 2022; Kong et al., 2019; Wang et al., 2020; Bruce et al., 2021). In particular, Thoker & Gall (2019) perform knowledge distillation without labels for the student modality data during training consistent with the UMA framework. Thus, their method is used as our student teacher baseline (ST). Nevertheless, their work is tested on one RGB+D dataset and lacks noise experiments, latent visuals, and comparisons to similar methods.

In addition, IMUTube (Kwon et al., 2020) and ChromoSim (Hao et al., 2022) simulate IMU data from videos to train an IMU model. In this instance, the teacher is the simulator, which trains the student IMU model without using any real IMU data. However, these approaches are resource-intensive, cannot easily extend to other modalities, and face simulation-reality gaps.

**Contrastive Alignment:** Unsupervised contrastive alignment methods for IMU data include IMU2CLIP (Moon et al., 2022), ImageBind (Girdhar et al., 2023), and mmg-Ego4d (Gong et al., 2023). While these approaches have shown promise in retrieval and generation tasks, they have limitations. All these methods focus exclusively on egocentric data. IMU2CLIP (Moon et al., 2022) aligns IMU data with text labels which violates the UMA setting. ImageBind (Girdhar et al., 2023) is not well tested with IMU data, is computationally intensive, and is not explicitly tested in the UMA setting. Mmg-Ego4d (Gong et al., 2023) addresses UMA with their zero-shot cross-modal transfer task, but their work is limited to a single private ego-centric dataset and focuses mainly on few-shot label learning. We use mmg-Ego4d as our CA baseline and run extensive UMA experiments, encompassing both egocentric and third-person camera-view public datasets Table 1. We further test its robustness to temporal distortions Table 2, its ability to scale with the latent vector size Figure 4, and we visualize its multimodal latent space Figure 6.

**Cross-modal Transfer Through Time:** Existing video understanding and robust sensor fusion methods often leverage temporal data, employing techniques such as temporal masking and reconstruction (Tong et al., 2022; Kong et al., 2019), spatio-temporal memory banks (Islam et al., 2022), or fusion of temporal chunks through transformer self-attention (Shi et al., 2024; Zhao et al., 2022; Wang et al., 2022). However, these approaches typically assume the availability of labeled data for all modalities in a specific task, limiting their applicability to the UMA setting. Unlike traditional

methods inspired by ViT (Arnab et al., 2021) and SwinTransformers (Liu et al., 2022), which chunk data into fixed or shifted time segments, C3T learns temporal tokens through temporal convolutions and uses them as a multi-vector latent space to align time-series modalities for cross-modal transfer.

## 6.2 BACKGROUND AND NOTATION

We investigate the creation of a robust multimodal latent space for human action recognition, denoted as $\mathcal{Z}$, that can be leveraged for UMA. In this context, robustness refers to the ability of the latent space to maintain consistent representations despite shifts in input distribution to different modalities or temporal noise. We assume that there exists a learnable projection $f^{(k)}$ from every modality $k \in 1, \ldots, M$ to this latent space $\mathcal{Z}$, i.e., $f^{(k)} : \mathcal{X}^{(k)} \rightarrow \mathcal{Z}$, such that the same actions viewed from different modalities map to proximal points in $\mathcal{Z}$, while distinct actions map to disparate regions. We further assume there exists a learnable mapping $h$ from the latent space $\mathcal{Z}$ to the label space of human actions $\mathcal{Y}$, i.e., $h : \mathcal{Z} \rightarrow \mathcal{Y}$. This mapping should be invariant to the originating modality of the latent representation. Our method leverages the intuition that proximity in the latent space $\mathcal{Z}$ indicates semantic similarity, i.e.neighboring points are likely map to the same class regardless of their originating modality. We use cosine similarity to measure *nearness*, as it's more effective in high-dimensional spaces than Euclidean distance (Radford et al., 2021). This choice focuses on directional similarity, mitigates the curse of dimensionality, and aligns with the distributional hypothesis in representation learning.

For simplicity, we experiment with two modalities, $M = 2$, and assume that a dataset of $n$ data points is split into four disjoint index sets $I_1 \cup I_2 \cup I_3 \cup I_4 \in \{1 \ldots n\}$. Under our UMA setting, during training, the model has access to two of these sub-datasets. One contains labeled data for one modality, $\mathcal{D}_{\text{HAR}} = \{(\mathbf{x}_i^{(1)}, \mathbf{y}_i)\}_{i=1}^{I_1}$, and the other contains synchronous data between the modalities, but these points are unlabeled, $\mathcal{D}_{\text{Align}} = \{(\mathbf{x}_i^{(1)}, \mathbf{x}_i^{(2)})\}_{i=1}^{I_2}$. This is analogous to having an existing sensor with labeled data and introducing a new sensor in which data can be synchronously collected, but there is no additional annotation effort (Figure 2). The third and fourth sets are used for validation and testing and contain only labeled data from the second modality, i.e. $\mathcal{D}_{\text{Val}} = \{(\mathbf{x}_i^{(2)}, \mathbf{y}_i)\}_{i=1}^{I_3}$ and $\mathcal{D}_{\text{Test}} = \{(\mathbf{x}_i^{(2)}, \mathbf{y}_i)\}_{i=1}^{I_4}$.

## 6.3 VISUALIZATIONS

Figure 7 is an expanded version of Figure 6 to show the points more clearly. Figure 8 visualizes the C3T latent space and a shifted input into that latent space. Figure 9 visualizes the C3T latent space with a shifted input and its original input. Figure 10 visualizes the CA latent space on a different dataset, CZU-MHAD. Figure 11 shows another visualization of the progression of the latent space in CA through training.

## 6.4 LIMITATIONS AND FUTURE DIRECTIONS:

Our novel visualizations of a joint multimodal latent space leave much room for future exploration. For example, in Figure 6 IMU data points consistently cluster towards the center of the plot, with RGB points surrounding them (more examples in Section 6.3). The consistency of this pattern suggests that the latent space may not be completely modality-agnostic. This phenomenon is an interesting direction for future research, potentially offering additional insights into cross-modal representations.

Furthermore, C3T's architecture has room to scale to larger models and backbones and test on other types of data. Given our theoretical intuition that alignment across local features is beneficial, and our time-shift visualizations in Figure 5 we believe C3T's method would generalize to other types of data or transfer scenarios involving time-series modalities.

We hypothesize C3T can easily extend to various tasks as well. Our training experiments in Section 4 show that C3T performs best by aligning the modalites first and our latent visualization in Figure 6 show structure in the multimodal latent space before labels are introduced. This suggest that our architecture may actually be able to transfer easily to various tasks. For example, after the alignment phase has been performed once, adding an additional task should only require training an additional task-specific head. We hypothesize that without additional alignment, we may be able to train a person identification task head-on the IMU latent representations and then perform person identification

with the RGB images without ever needing additional RGB data. This sort of multi-task transfer investigation would be a valuable future extension of this work.

**Discussion of potential positive and negative impacts of the work:** As stated in the introduction, cross-modal transfer for sensor data has numerous positive applications, in healthcare, smart homes, smart devices, user interface design etc. Some negative impacts that may occur are that social bias may transfer across modalities, certain modalities might not be accessible to individuals with disabilities, exacerbating the technology gap between the able-bodied and disabled, and cross-modal transfer may increase certain security and privacy concerns if someone's identity/information can easily be transferred across modalities.

**LLM Use** LLMs were used only to rephrase writing to help with grammar, clarity, and flow. They were also used for latex syntax and formatting assistance.

## 6.5 ADDITIONAL EXPERIMENTS

*Extended: Can UMA methods retain performance on the labeled modality? Can they leverage both modalities?*

**Experimental setup:** Table 5 shows the result of training in the UMA setting but testing with all combinations of the modalities. Any inputs can be used to perform HAR by simply using the HAR module $h$ on an estimate for the latent vector, $\hat{z}$ derived from the modalities. For ST, $h$ can be viewed as the identity, and $z$'s are the output logits.

1. **RGB (Supervised Learning):** Tests the model on RGB data, which was labeled during training, thus this is supervised paradigm. The estimated latent vector is given by $f^{(1)}(x_i^{(1)}) = \hat{z}_i$.
2. **IMU (UMA):** Is the main result of this paper and described above in Section 2. Here the estimated latent vector is given by $f^{(2)}(x_i^{(2)}) = \hat{z}_i$.
3. **Both (Sensor Fusion):** Merges latent vectors from each modality. Assuming each estimate is equally as good as the other: $\hat{z}_i = \mathbb{E}[z_i|x_i^{(1)}, x_i^{(2)}] = E[z_i|\hat{z}_i^{(1)}, \hat{z}_i^{(2)}] = \frac{\hat{z}_i^{(1)}+\hat{z}_i^{(2)}}{|\hat{z}_i^{(1)}+\hat{z}_i^{(2)}|}$ . Given that we align the latent vectors from different modalities by minimizing the angle between them, i.e. cosine similarity, we also fuse vectors by generating the normalized vector whose angle is halfway between the estimated vectors.

**Results:** As expected, Table 5 shows that C3T outperforms the other methods. When comparing the performances in the different test scenarios, our experiments indicate that when given both modalities, fusion performs better than the RGB model alone. Instead of introducing noise or uncertainty into the model, introducing an unlabeled modality may add structure to the shared latent space that bolsters performance, especially if that modality is highly informative for the given task. This observation bears resemblance to knowledge distillation methods, where an auxiliary modality during training leads to improved testing outcomes, however, these methods usually assume that auxiliary modality is labeled (Chen et al., 2023). The application of UMA to sensor fusion presents a promising avenue for future research.

**Additional Dataset:** We revisited this question of testing on different modalities from the main paper and attempted to test on other modalities for different datasets. We noticed that the results were consistent with our main findings, that often testing with both modalities performs better than testing with the training modality that had labels (RGB)! See Table 6 below for the results on the MMACT dataset.

Table 6: **UMA Testing on Each Modality** Accuracy MMACT (Kong et al., 2019) when trained for UMA and tested on only RGB data, only IMU data, or Both.

| Model | 1. RGB | 2. IMU | 3. Both |
|-------|--------|--------|---------|
| ST    | 25.8%  | 16.7%  | **25.8%** |
| CA    | 39.9%  | 26.9%  | **41.3%** |
| C3T   | 39.2%  | 31.7%  | **47.3%** |

*How quickly can our models learn from labeled IMU samples?*

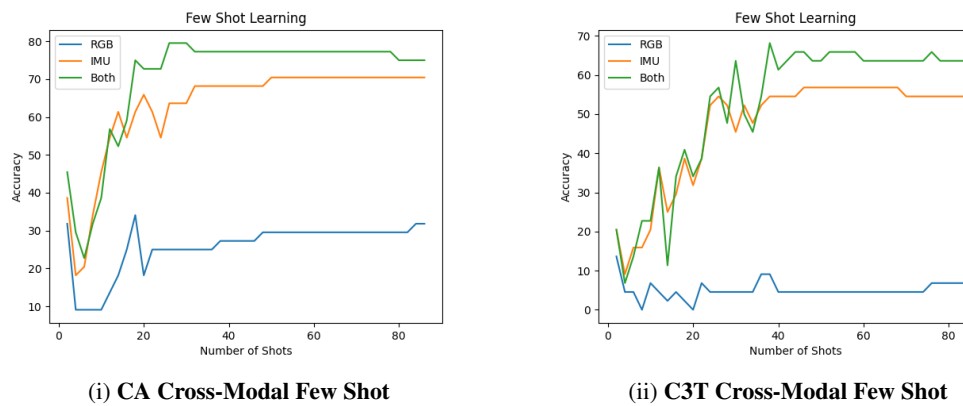

(i) **CA Cross-Modal Few Shot**       (ii) **C3T Cross-Modal Few Shot**

Figure 13: **Cross-Modal Few Shot Learning Comparison:** (a) CA method and (b) C3T method performance in cross-modal few shot learning scenarios when testing on IMU, RGB and Both modalities. RGB performance remains the same because the learning shots only contain labeled IMU data. However, we can see IMU and Both performance rise.

As illustrated in Figure 13, CA demonstrates faster learning, reaching peak performance within 20 shots, while C3T requires about 40 shots. Neither method matches the supervised IMU performance of 87.9% reported in Table 1, but they approach the fusion performance of approximately 62%.

It's important to note that this comparison with the supervised baselines may not be entirely fair, as the supervised baselines had access to the entire Train HAR dataset (40% of the data), whereas, the few-shot learning was conducted on the validation set (10% of the data). Given that the supervised IMU and fusion models share the same architecture as CA, repurposed for the supervised setting, we would expect similar performance under equal conditions.

Nonetheless, these experiments clearly demonstrate CA's superior ability in few-shot cross-modal learning compared to C3T. As shown in Figure 13, both IMU and combined modality performance improve with increasing shots, while RGB performance remains constant due to the learning shots containing only labeled IMU data.

### 6.6 ADDITIONAL ABLATIONS:

We conducted a brief ablation study on the alignment loss, comparing our cosine similarity approach with the conventional L2 loss. As shown in Table 7, the results strongly support our initial intuition presented in Section 6.2. The substantial performance gap between cosine similarity and L2 loss for both CA and C3T models underscores that cosine similarity is indeed a more effective measure of alignment for high-dimensional vectors in this context. These findings align with well-established principles in high-dimensional space analysis, reinforcing the validity of our approach. Given the well-established nature of these results, we have included this comparison in the supplementary material rather than the main paper, focusing the primary discussion on novel contributions.

Table 7: **Alignment Loss Comparison:** Performance of CA and C3T models using Cosine Similarity and L2 loss for alignment on the UTD-MHAD dataset.

| Model | Cosine Similarity | L2 Loss |
|-------|-------------------|---------|
| CA    | **44.32**         | 2.27    |
| C3T   | **62.50**         | 3.41    |

Lastly, we provide an extension of our architecture ablations in Table 8. In this extension, we also ablate the encoders of the supervised baselines, which are the same modules used for ST, CA and C3T, however, trained in the supervised setting, as opposed to UMA. The only interesting result is

Table 8: **Architecture Ablation Extensions:** This shows an extension of our architecture ablations (Table 3) to baselines. It shows a comparison of different architectures for RGB and IMU encoders across various methods. We report encoder types for spatial and temporal dimensions of RGB data, and the temporal dimension for IMU data, along with the number of parameters (in millions) and accuracy for each configuration. Convolutional architectures generally yielded superior performance, while still maintaining a relatively low model size. These results indicate that C3T's performance advantage stems from its methodological approach rather than solely from it's attention head or size.

| Method | RGB | | IMU | Params (M) | Accuracy (%) |
| | Spatial | Temporal | Temporal | | |
|---|---|---|---|---|---|
| ST | Conv | Conv | Conv | 129.2 | **12.9** |
| | Conv | Conv | Attn | 97.8 | 10.2 |
| | Conv | Attn | Conv | 871.2 | 11.4 |
| | Attn | Conv | Conv | 291.5 | 5.7 |
| CA | Conv | Conv | Conv | 163.8 | **38.6** |
| | Conv | Conv | Attn | 132.3 | 19.3 |
| | Conv | Attn | Conv | 905.7 | 34.1 |
| | Attn | Conv | Conv | 326.0 | 26.1 |
| C3T | Conv | Conv | Conv | 137.7 | **62.5** |
| | Conv | Conv | Attn | 106.3 | 15.9 |
| | Conv | Attn | Conv | 879.6 | 53.4 |
| | Attn | Conv | Conv | 300.0 | 33.0 |
| FUSION | Conv | Conv | Conv | 163.8 | 62.5 |
| | Conv | Conv | Attn | 132.3 | 77.3 |
| | Conv | Attn | Conv | 905.7 | **89.8** |
| | Attn | Conv | Conv | 326.0 | 64.8 |
| IMU | – | – | Conv | 32.0 | **87.9** |
| | – | – | Attn | 0.5 | 27.3 |
| RGB | Conv | Conv | – | 97.3 | 53.8 |
| | Conv | Attn | – | 839.2 | **71.6** |
| | Attn | Conv | – | 259.5 | 64.8 |

that the RGB baseline may have performed better with an attention-based temporal feature extractor. This could potentially imply that the C3T does not surpass the best supervised RGB model, as we noted in the main results. It is not necessary to ablate the unimodal supervised setting because that is not the main novel method of this work. We believe there exist other Video Action recognition models that are better. However, this does not contradict the main findings of this paper that C3T is the most robust to temporal noise in the UMA cross-modal transfer setting.

### 6.7 METHODS:

Table 9: **Data Splits for Unsupervised Modality Adaptation (UMA):** During training, no labeled IMU data is present, thus the model can only leverage the correlations between $X^{\text{IMU}}$ and $X^{\text{RGB}}$ to learn classes for IMU data.

| Split | $X^{\text{RGB}}$ | $X^{\text{IMU}}$ | $Y$ | % of Data |
|---|---|---|---|---|
| $\mathcal{D}_{\text{HAR}}$: Train a) | ✓ | | ✓ | 40% |
| $\mathcal{D}_{\text{Align}}$: Train b) | ✓ | ✓ | | 40% |
| $\mathcal{D}_{\text{Val}}$: Val | | ✓ | ✓ | 10% |
| $\mathcal{D}_{\text{Test}}$: Test | | ✓ | ✓ | 10% |

Table 9 shows the data splits used for Unsupervised Modality Adaptation (UMA) training. All tables report the accuracy on the $\mathcal{D}_{Test}$ for each method ( accuracy $= \frac{1}{I_4} \sum_{i=1}^{I_4} \mathbb{1}_{\hat{\mathbf{y}}_i = \mathbf{y}_i}$).

Table 10: **UMA performance compared to supervised baselines Using Train Method 1:** Each method is modular and can be decomposed to perform in the traditional supervised setting, or can be combined into ST, CA or CT3 to perform UMA. We developed all models from scratch, however, ST and CA resemble existing methods whereas C3T introduces novel mechanisms. Top-1 and Top-3 accuracies are reported for each dataset. Although ST performs poorly, it performs significantly better than randomly guessing, indicating it is still learning action information from the RGB data without any labels. Please note the CA row of this table reports the results when trained using training method 1 (align first). The overall trends do not differ from Table 1, indicating that the performance results of C3T are not biased by the training method.

| | Model | UTD-MHAD | | CZU-MHAD | | MMACT | | MMEA-CL | |
|---|---|---|---|---|---|---|---|---|---|
| | | Top-1 | Top-3 | Top-1 | Top-3 | Top-1 | Top-3 | Top-1 | Top-3 |
| Supervised | IMU | **87.9** | **97.7** | **95.1** | 98.2 | 70.0 | 90.0 | 65.8 | 87.6 |
| | RGB | 53.8 | 73.1 | 94.0 | **99.7** | 42.1 | 61.6 | 54.2 | 77.1 |
| | Fusion | 62.5 | 82.2 | 95.0 | 98.5 | **76.7** | **92.0** | **80.1** | **92.7** |
| UMA | Random | 3.7 | 11.1 | 4.6 | 16.6 | 2.9 | 8.6 | 3.1 | 9.4 |
| | ST | 12.9 | 24.6 | 41.1 | 61.9 | 17.6 | 34.7 | 9.9 | 22.7 |
| | CA | 38.6 | 56.1 | 81.0 | 95.5 | 27.3 | 45.6 | 42.3 | 62.1 |
| | C3T | **62.5** | **86.4** | **84.2** | **96.7** | **32.4** | **57.9** | **51.2** | **78.8** |

For reproducibility and to guarantee the scientific rigor of our experiments, our main results table in Table 1 was run 3 times with random seeds pytorch 1,2 and 3. The full results are given in Figure 12. The results report the average values across all trials and their standard deviations across those 3 trials. We note, that although the standard deviations are quite large, the difference in performance between the methods is typically larger than one standard deviation, implying statistical significance in our comparative analysis. This suggests that the observed performance improvements of C3T over baseline methods are not merely due to random variation but represent genuine algorithmic advantages. Furthermore, the consistent pattern of outperformance across multiple datasets strengthens the reliability of our conclusions, as the probability of observing such consistent results by chance across independent experiments would be exceedingly low. We also note that the supervised methods were only trained on 40% of the data given in $\mathcal{D}_{\text{HAR}}$ for a fair comparison to the other methods.

**Datasets and Hyperparameters - Additional Info:** We present results on four diverse datasets: (1) UTD-MHAD (Chen et al., 2015), a small yet structured dataset; (2) CZU-MHAD (Chao et al., 2022), a slightly larger dataset captured in a controlled environment; (3) MMACT (Kong et al., 2019), a very large dataset with various challenges including different view-angles, scenes, and occlusions; and (4) MMEA-CL (Xu et al., 2023), an egocentric camera dataset. For each dataset, we create an approximately 40-40-10-10 percent data split for the $\mathcal{D}_{\text{Align}}$, $\mathcal{D}_{\text{HAR}}$, $\mathcal{D}_{\text{Val}}$, and $\mathcal{D}_{\text{Test}}$ sets respectively, as shown in Appendix Table 9. $\mathcal{D}_{\text{Val}}$ was used to perform a minor hyperparameter search on the UTD-MHAD dataset. The methods performed best with a learning rate of $1.5 \times 10^{-4}$, a batch size of 16, num_workers of 4, and a latent representation dimension of 2048 with an Adam optimizer. The preprocessing steps downsample the video to $t = 30$ frames, and C3T extracts $t_{\text{fm}} = 15$ latent vectors per sample. Experiments were implemented in Pytorch and run on a 16GB NVIDIA Quadro RTX 5000, four 48GB A40s, or four 48GB A100s and each run took between 4-10 hours, depending on the dataset and GPU availability. More detailed information about each dataset and implementation can be found in Appendix Section 6.8.

**Main Table With All Train Method 1:** Before the ablations discovered that Train method 4 (combined loss) was better for CA we used method 1 (Align first). Table 10 shows the original experiments with all UMA models trained using method 1 (align first) and we observe no difference in the resulting rankings of the method.

**Student Teacher:** Below is the standard cross-entropy loss that was employed for the student teacher methods.

$$\mathcal{L}_{CE}(P_{\hat{y}}, P_y) = -\frac{1}{N} \sum_{i=1}^{N} \sum_{j=1}^{C} \mathbb{1}_{y_i=j} \log\left(\frac{\exp \hat{y}_{i,j}}{\sum_{i=1}^{M} \exp \hat{y}_{i,j}}\right) \tag{3}$$

where $\hat{y}_i$ is the output of the $i$th sample in the batch of $N$ samples, $\hat{y}_{i,j}$ is the score for the $j$th class out of $C$ classes, and $P_y$ represents the probability distribution produced by a given model's output logits. The teacher network minimizes $\mathcal{L}_{CE}(P_{f^1(x)}, P_y)$ and the student minimizes $\mathcal{L}_{CE}(P_{f^2(x)}, P_{f^2(x)})$. Since the student approximates the teacher and the teacher approximates the true distribution, this implies that the student can only be as good as the teacher at approximating the true distribution:

$$\mathcal{L}_{CE}(P_{f^1(x)}, P_y) \leq \mathcal{L}_{CE}(P_{f^2(x)}, P_y) \tag{4}$$

**C3T Modules** Here we provide a more precise formulation of the modules used for C3T.

The updated modules are as follows:
**Video Feature Encoder** $f^{(1)} : \mathcal{X}^{(1)} \to \mathcal{Z}^{t_{rec}}$: This module applies a pretrained Resnet18 to every frame a video and then performs a single 3D convolution. The resulting output is a set $t_{rec}$ $z$: $\hat{\mathbf{Z}}^{(1)} = (\hat{z}_1^{(1)} \dots \hat{z}_{t_{rec}}^{(1)})$.
**IMU Feature Encoder** $f^{(2)} : \mathcal{X}^{(2)} \to \mathcal{Z}^{t_{rec}}$: This is a 1D CNN that decreases the time dimension to $t_{rec}$, resulting in an output of $\hat{\mathbf{Z}}^{(2)} = (\hat{z}_1^{(2)} \dots \hat{z}_{t_{rec}}^{(2)})$.
**HAR Task Decoder** $h : \mathcal{Z}^{t_{rec}} \to \mathcal{Y}$: This module is like a transformer encoder that uses self-attention on an input sequence of length $t_{rec}$ vectors appended with a learned class token. The output class token of the self attention layer is then passed through a FFN and outputs a single action label $y_i$.

## 6.8 DATASETS

Here we provide more information on the datasets and how they were used in our experiments.

**UTD-MHAD** Most of the development and experiments were performed on the UTD-Multi-modal Human Action Dataset (UTD-MHAD) (Chen et al., 2015). This dataset consists of roughly 861 sequences of RGB, skeletal, depth and an inertial sensor, with 27 different labeled action classes performed by 8 subjects 4 times. The inertial sensor provided 3-axis acceleration and 3-axis gyroscopic information, and all 6 channels were used for in our model as the IMU input. Given our motivation, we only use the video and inertial data; however, CA can easily be extended to multiple modalities.

**CZU-MHAD** The Changzhhou MHAD (Chao et al., 2022) dataset provides about 1,170 sequences and includes depth information from a Kinect camera synchronized with 10 IMU sensors, each 6 channels, in a very controlled setting with a user directly facing the camera for 22 actions. They do not provide RGB information, thus we use depth as the visual modality, broadcast it to 3 channels, and pass it into the RGB module. We concatenate the IMU data to provide a 60-channel input as the IMU modality and use depth as the input modality. Given the controlled environment and dense IMU streams, the models performed the best on this dataset.

**MMACT** The MMAct dataset (Kong et al., 2019) is a large scale dataset containing about 1,900 sequences of 35 action classes from 40 subjects on 7 modalities. This data is challenging because it provides data from 5 different scenes, including sitting a desk, or performing an action that is partially occluded by an object. Furthermore, the data was collected with the user facing random angles at random times. The dataset contains 4 different cameras at 4 corners of the room, and it measures acceleration on the user's watch and acceleration, gyroscope and orientation data from a user's phone in their pocket. We only use the cross-view camera 1 data, and again we concatenate the four 3-axis inertial sensors into one 12-channel IMU modality.

**MMEA-CL** The Multi-Modal Egocentric Activity recognition dataset for Continual Learning (MMEA-CL) (Xu et al., 2023) is a recent dataset motivated by learning strong visual-IMU based representations that can be used for continual learning. It provides about 6,4000 samples of synchronized first-person video clips and 6-channel accelerometer and gyroscope data from a wrist worn IMU for 32 classes. The dataset's labels feature realistic daily actions in the wild, as opposed to recorded sequences in a lab. Due to issues with the data and technical constraints, we downsize the data proportionally from each class and use about the first 1,000 samples. However, CT3's superior performance shows how this method can generalize to a camera view (ego-centric camera), and different types of activities.

Table 11: **UMA with Existing Methods:** Most methods fail to adapt to zero-shot cross-modal transfer from the RGB to IMU sensor modalities. Imagebind performs well on MMEA, which is an eogecentric dataset, similar to Ego-4d in which Imagebind was trained on.

| Model | UTD-MHAD | MMACT | MMEA- CL | CZU-MHAD |
|---|---|---|---|---|
| Sensor Fusion (2019) (Wei et al., 2019) | 5.2% | 3.2 % | 4.1 % | 4.5 % |
| HAMLET (2020) (Islam & Iqbal, 2020) | 4.6 % | 3.2 % | 4.1 % | 4.5 % |
| ImageBind (2023) (Girdhar et al., 2023) | 11.3 % | 4.6 % | 40.1 % | 4.54 % |
| Student Teacher | 12.9 % | 17.6 % | 9.9 % | 41.1% |
| Contrastive Alignement | 38.6 % | 27.3 % | 42.3 % | 81.0 % |
| Cross-modal Transfer Through Time (Ours) | **62.5** % | **32.4** % | **51.2** % | **84.2** % |

### 6.9 BASELINES

This method attempts to adapt existing methods to UMA and compare them as baselines against our methods.

Many works deal with robustness to missing sensor data during training or testing, however, few works deal with zero-labeled training data from one modality. As a result, constructing baselines is tricky and most methods had to be modified or adapted to fit our our approach. Even so, as shown in Table 11 These methods perform very poorly in the UMA setting.

We would like to note that all the baselines and methods were trained and tested on the same data splits, i.e. it's not the case that they have different train a) and train b) splits. We believe that this allows for fair comparison. In addition, the supervised baselines were only trained on the Train a) Supervised HAR split. This ensures that the supervised baselines do not have an unfair advantage of seeing more data (they also only see 40% of the labeled data not 80%). The exact data splits are also provided in a separate repository linked in our code.

#### 6.9.1 SENSOR FUSION BASELINES

Sensor fusion is often broken down into the following 3 methods based on where the data are combined (Ramachandram & Taylor, 2017; Majumder & Kehtarnavaz, 2020; Sharma et al., 1998), also shown in Figure 14:

1. Early or data-level fusion combines the raw sensor outputs before any processing.
2. Middle/intermediate or feature-level fusion combines each sensor modality after some preprocessing or feature extraction.
3. Late or decision-level fusion combines the raw output, essentially ensembling separate models.

Many IMU-RGB based sensor fusion models have the ability to train on partially available or corrupted data and are robust to missing modalities during inference (Islam et al., 2022; Islam & Iqbal, 2020). Sensor fusion works rarely attempt the extreme case where one modality is completely unlabeled during training. Existing sensor fusion methods can be adapted to our setup using a psuedo- labeling technique, similar to the student-teacher model above. The difference for sensor fusion is that the model learns a joint distribution between the two modalities as opposed to two separate distributions. Thus the model may be able to learn some correlation between the modalities. Nonetheless, we show that these methods cannot generalize to the scenario where there is zero-labeled training data for one modality.

Let $g(\cdot, \cdot) : (\mathcal{X}^{(1)}, \mathcal{X}^{(2)}) \to \mathcal{Y}$. Our approach uses $\mathcal{D}_{HAR}$, to train by passing in zeros for one modality, e.g. we train $g(\cdot, \mathbf{0}) : \mathcal{X}^{(1)} \to \mathcal{Y}$. Then, with $\mathcal{D}_{Align}$ we use $g(\cdot, \mathbf{0})$ to generated psuedo-labels and then train $g(\mathbf{0}, \cdot,)$ with those labels.

We reproduced the conventional sensor fusion models (early, feature, and late) from (Wei et al., 2019) and indicate the performance of the top model on 11. We further reproduce a self-attention based sensor fusion appraoch (HAMLET (Islam & Iqbal, 2020)) and tested it on our setup. We follow a very similar architecture; however, extract spatio-temporal results using 3D convolution in the video as opposed to an LSTM. This method provides similar results as the LSTM method on the standard sensor fusion problem. To verify the integrity of our reproduced models we compared to state-of-the-art reported methods and showed similar performance results. The results are given in Table 12. We selected HAMLET due to its state-of-the-art performance on the UTD-MHAD dataset, making it an ideal benchmarks for comparison with our model. These experiments prove

that our reproduced baselines are comparable to SOTA method. In addition, these baselines fail to perform well in the UMA setting underscoring the importance and novelty of our work.

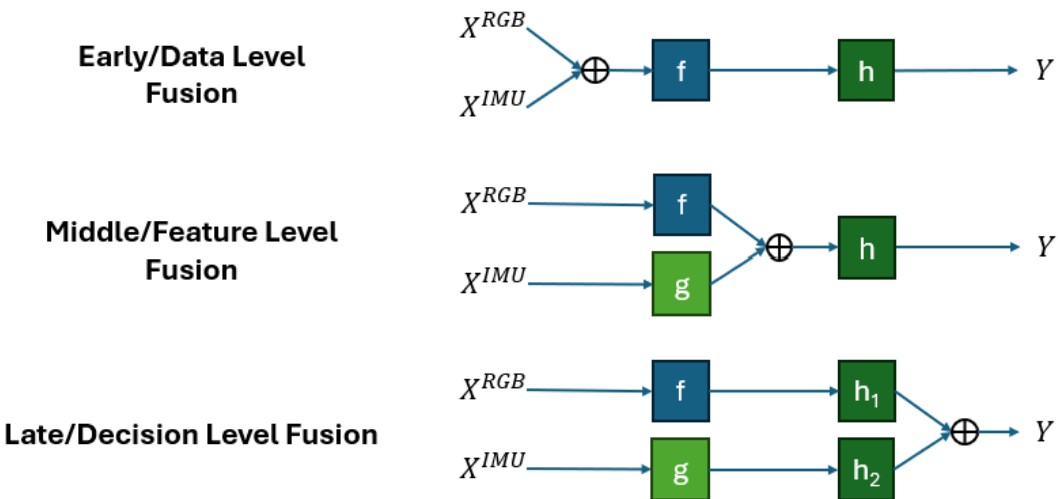

Figure 14: Types of Sensor Fusion

Table 12: SOTA Sensor Fusion Performance on UTD-MHAD. † (Wei et al., 2019), ∗ (Islam & Iqbal, 2020)

| Reported Models | Accuracy |
|---|---|
| HAMLET ∗ | 95.12% |
| Wei et al. † | 95.6% |
| Reproduced from † | Accuracy |
| Early Fusion | 86.71% |
| Feature Fusion | 95.60% |
| Late Fusion | 94.22% |

### 6.9.2 Contrastive Learning Baseline

ImageBind (Girdhar et al., 2023) learns encoders for 6 modalities, (Images/Videos, Text, Audio, Depth, Thermal and IMU) by performing CLIP's training method (Radford et al., 2021) between each of those encoders and the Image/Video encoder. It was well tested for image, text and audio-based alignment, retrieval, and latent space generation tasks, however was not well tested with IMU data and not used for specific tasks, such as HAR. In addition, one fundamental difference between Imagebind and CA is that Imagebind constructs a latent space amongst the sensing modalities and text and aligns between them. We hypothesize that this vector space is more difficult and unnecessary to construct, for human action recognition using sensing modalities. The text modality, although sequential in nature, does not have a time dimension, thus it cannot leverage correlations between modalities in time like C3T. We work with the original Imagebind model and code released on github [1]. We perform two conventional task-specific adaptations for CLIP models: Zero Shot transfer and Linear Probing:

**Zero-Shot Transfer:** Let's denote the video, IMU and text encoders as $g^{(1)} : \mathcal{X}^{(1)} \to \mathcal{Z}, g^{(2)} : \mathcal{X}^{(2)} \to \mathcal{Z}$, and $g^{(3)} : \mathcal{X}^{(3)} \to \mathcal{Z}$ respectively. First, we attempt zero-shot transfer. Here, we pass all the action labels through the text encoder of the pretrained model. For a dataset with $C$ classes, we have $\hat{Z}^{(3)} = (\hat{z}_1^{(3)} \dots \hat{z}_C^{(3)})$. Finally, for a given IMU sample $(x_i^{(2)}, y_i) \in \mathcal{D}_{Test}$, we pass $x_i^{(2)}$ through the IMU encoder $g^{(2)}$ and retrieve $\hat{z}^{(2)}$. Then, we classify the point by looking at which points gives the highest cosine similarity score in the latent space, e.g. $\hat{y}_i = argmax_j \frac{\langle x_i^{(2)}, \hat{z}_j^{(3)} \rangle}{||\langle x_i^{(2)}, \hat{z}_j^{(3)} \rangle||}$. The video encoder $g^{(1)}$ goes unused as no training was done, and in UMA, there is no video data during testing. This performed poorly and the results are not reported.

---

[1] https://github.com/facebookresearch/ImageBind

**Linear Probing:** Given that ImageBind is a large model trained on massive corpuses of data it becomes impractical to train the model from scratch on our smaller datasets collected from wearables and edge devices. Instead, we fine-tuned the ImageBind model using a linear projection head on the encoders, that can then be trained for a specific task. The results of this method are depicted in Table 11.

The results show a poor generalization of Imagebind to most experiments on our setup, and we hypothesize a few reasons. Firstly, ImageBind is a large model and may either overfit to small datasets, or not have enough training examples to learn strong enough representations. Second, ImageBind was pre-trained on Ego4D and Aria which contain egocentric videos to align noisy captions with the IMU data, whereas our datasets had fixed labels and were mostly 3rd person perspective. In fact ImageBind performed the best on the one egocentric dataset we used, MMEA-CL(Xu et al., 2023). Lastly, Imagebind was trained on a IMU sequences of 10s length sampled at a much higher frequency, thus we zero-padded or upsampled the IMU data to fit into ImageBind's IMU encoder, and the sparse or repetitive signal may have been too weak for ImageBind's encoder to accurately interpret the data.

