# OpenReview forum: "Cross-modal Transfer Through Time for Human Activity Recogntion"
_ICLR.cc/2026/Conference — ICLR 2026 Conference Withdrawn Submission_

### Official Review · Reviewer_bgCj · 2025-10-27

**Soundness:** 2
**Presentation:** 2
**Contribution:** 2
**Rating:** 4
**Confidence:** 3

**Summary:**

This paper introduces Cross-modal Transfer Through Time (C3T) for Unsupervised Modality Adaptation (UMA) in Human Activity Recognition (HAR), focusing on transferring knowledge from labeled RGB videos to unlabeled IMU data. Existing methods compress time-series into single latent vectors, losing temporal details; C3T aligns temporal latent vectors from convolution feature maps using an extended contrastive loss (LC3T), followed by self-attention for classification.

**Strengths:**

1. **Relevant Problem and Innovation**: The UMA setting is practical for unlabeled sensors in wearables/IoT, addressing real-world challenges like labeling scarcity and temporal variations.
2. **Empirical Rigor**: Comprehensive evaluations on diverse datasets (UTD-MHAD, CZU-MHAD, MMACT, MMEA-CL) demonstrate consistent outperformance of baselines, with ablations on latent size and noise.
3. **Potential Impact**: Opens avenues for generalizable time-series models in healthcare, smart homes, and HCI, with implications for emerging modalities beyond RGB-IMU.

**Weaknesses:**

1. **Limited Novelty**: While effective, C3T feels incremental. Temporal alignment echoes recent frameworks like COMODO, and multi-modal contrastive learning extensions (e.g., CoMM). The paper cites a 2024 IMU cross-modal survey but overlooks broader 2024-2025 surveys on generalizable HAR and multi-modal CL advances, potentially understating similarities to sequence-based CL in video/HAR. Baselines are appropriate but not SOTA.
2. **Scope and Assumptions**: Relies on paired RGB-IMU data; no handling of unpaired scenarios common in real-world settings. Limited to RGB-to-IMU; lacks tests on other pairs (e.g., WiFi, radar). Datasets are small/old.
3. **Analytical Depth**: No theoretical justification for LC3T's robustness (e.g., invariance bounds). Privacy/ethics in sensor data ignored. Broader impacts mentioned but not explored deeply.

**Questions:**

See above.

---

> ### Author Response · Authors · 2025-12-03
> **Response to bgCj Part 1**
>
> We appreciate Revierwer bgCj’s feedback and thoughts. They highlight the relevant and innovative problem formulation and potential impact, as well as the empirical rigor of our experiments.
>
> We hope to clarify the novelty and justify the scope and analytical depth of our method below:
>
> ## Response to Weaknesses
> ### **1. Limited Novelty**:
> Please review the "Rebuttal Overview" and the "Response to Reviewer iCut Part 1" for clarification on our novelty. We will continue here responding to the reviewers' other concerns and clarifying how C3T differs from the works they provide.
>
> This design differs from sequence-based contrastive learning and temporal attention in video/HAR in two key ways:
> - Prior multimodal contrastive works (e.g., IMU2CLIP, ImageBind, mmg-Ego4D) align global embeddings (a single embedding for the entire video) and are not evaluated in a strict zero-label target-modality UMA regime.​
> - Temporal transformers (e.g., ViViT-style) use attention within a modality to model time, but do not perform alignment across modalities at the temporal feature level, nor do they tackle UMA where labels exist only in one modality.​
>
> Regarding the specifically mentioned works:
> ### COMODO [1]
> COMODO is conceptually much closer to standard contrastive alignment (like our CA baseline, IMU2CLIP, mmg-Ego4D, and ImageBind) than to C3T. Its primary innovation is a FIFO queue mechanism that maintains a dynamic repository of video-level embeddings from the previous N videos, allowing each IMU sequence to align not only with its current RGB positive pair but also with negatives/positives drawn from this temporal queue of distinct videos. Critically:​
> - COMODO still operates on global embeddings (one vector per video/IMU sequence), not sets of fine-grained temporal tokens from convolutional feature maps.
> - The queue provides cross-video negatives, not alignment across time-steps within the same video—the core of C3T's LC3T loss, which explicitly pulls corresponding temporal positions together across modalities (Eq. 2).​
>
> Thus, COMODO is orthogonal to C3T: its queue could be layered atop C3T by storing queues of temporal token sets rather than single vectors per video. Moreover, COMODO is not evaluated in strict UMA: while "label-free" during alignment, inference requires IMU labels to train a downstream classifier (e.g., "class labels only used to train SVM on extracted features," page 4 of [1], bottom of Section 3.2 Inference). It extracts IMU features but does not transfer a shared task head zero-shot from RGB to IMU, as C3T does.​
>
> Finally, COMODO lacks the comprehensive analysis we provide: no temporal robustness experiments under cropping/misalignment/dilation (Table 2), no scaling studies (Fig. 4), and no latent-space visualizations showing multimodal cluster formation or attention shifts (Figs. 5–6). The only similarity between COMODO and C3T is that they test with similar modalities and have similar motivations (transfer from Video to IMU for HAR); however, they are fundamentally different ideas and tested and observed in different problem setups.

---

> ### Author Response · Authors · 2025-12-03
> **Response to bgCj Part 2**
>
> ### CoMM [2]
> CoMM is an extension of basic multimodal contrastive learning, but its design and goals differ fundamentally from C3T and from the UMA setting we study. In CoMM, the core idea is to first fuse modalities into a single multimodal representation and then maximize mutual information between augmented versions of that multimodal feature. The objective is to decompose multimodal information into redundancy, uniqueness, and synergy terms and to learn task-agnostic multimodal features that capture these interactions in a global joint space. Concretely:​
> - CoMM operates on multimodal feature vectors that already combine all modalities; it does not perform alignment between separate unimodal encoders at the temporal level.
> - The “multiple vectors” produced in CoMM correspond to augmented or factorized views of the multimodal representation (e.g., synergy/uniqueness/redundancy), not to time-indexed tokens of a video or time series. The contrastive objective is applied between augmented multimodal features, not across per-timestep latent vectors from different modalities.
> - CoMM is designed as a self-supervised pretraining framework for general multimodal tasks, followed by supervised evaluation (linear probing, fine-tuning) on downstream tasks with labels in the target modalities. It does not address zero-shot transfer of a task head from one modality to another under strict UMA (no labels in the target modality).​
>
> By contrast, C3T:
> - Keeps separate unimodal encoders and performs token-level alignment across modalities in time, aligning fine-grained temporal latent vectors from RGB and IMU rather than a single fused embedding.
> - Directly targets UMA for RGB→IMU HAR, where the classifier is trained only on RGB labels and then used as-is on aligned IMU representations, without any IMU labels during training.​
> - Focuses on temporal robustness and cross-modal transfer (e.g., to time shift, misalignment, dilation), rather than on decomposing multimodal information into redundancy/uniqueness/synergy or modeling arbitrary numbers of modalities.
>
> In that sense, CoMM’s innovation is adjacent but not overlapping with C3T’s. One could, in principle, replace the standard CA loss inside C3T with a CoMM-style contrastive objective defined on each temporal token time step, but that would be an additional design choice not a reiteration of C3T. Moreover, CoMM requires labeled evaluation in each modality (e.g., linear probing) and is not demonstrated to transfer a task head zero-shot from one modality to another, especially not in RGB–IMU video/sensor settings.
>
> Finally, CoMM’s empirical focus is on broad, task-agnostic multimodal representation learning across many domains (images, text, audio, tabular, etc.), whereas our work provides depth on temporal alignment for sensor-based HAR: temporal noise experiments (Table 2), scaling behavior (Fig. 4), detailed multimodal latent-space visualizations, and UMA-specific transfer and fusion experiments. We will clarify these distinctions in the revised related work section, emphasizing that CoMM and C3T address complementary questions and that CoMM is not a direct baseline for the strict UMA, RGB→IMU, temporally-aligned transfer setting studied in our paper.
>
> **Regarding the comment that we “overlook broader 2024–2025 surveys on generalizable HAR and multi-modal CL advances,”** our intention was not to restrict the related work to hyper-focused surveys, but to foreground those most directly aligned with cross-modal transfer between time series sensor modalities. Work from 2024 remains recent in ML-for-sensor-data research, a field slightly slower-paced than vision-language models. Many multimodal CL surveys primarily cover image–text or other discrete-modality scenarios, and their assumptions and objectives (e.g., large-scale vision–language pretraining) are not directly related to the temporal, sequence-level issues that C3T is designed to address. In the revised version, we can explicitly position C3T with respect to these broader trends. If there are specific survey papers the reviewer had in mind, we would be grateful for pointers so that we can cite and contrast them accurately.

---

> ### Author Response · Authors · 2025-12-03
> **Response to bgCj Part 3**
>
> ### **2. Scope and Assumptions**:
> **Paired vs. unpaired data.** Our UMA setting deliberately assumes access to synchronous but unlabeled RGB–IMU pairs for alignment (DAlign), which is realistic in many deployments (e.g., a smart home camera and a smartwatch worn simultaneously) and allows learning a shared latent space without costly IMU annotation. Unpaired scenarios are indeed common, and handling them would likely require additional machinery (e.g., cycle-consistency, generative alignment, or weak synchronization cues) that is orthogonal to the core UMA contribution. We agree this is an important extension and will state more explicitly in the Limitations section that C3T currently targets the paired-but-unlabeled regime, and that extending UMA to unpaired multimodal data is a promising direction for future work.​
>
> **Other modality pairs.** We chose RGB→IMU as our primary testbed because (i) it is central for wearables and smart environments, and (ii) multiple public RGB–IMU datasets are available, enabling a unified UMA protocol and controlled comparisons. The LC3T formulation itself is modality-agnostic; given synchronized data, it applies equally to other time-series modalities (e.g., WiFi, radar, EMG). Furthermore, all our intuition and analysis was focused on the temporal aspect of the data, not any unique characteristics of IMU or RGB data (e.g., Table 2 noise experiments, Figure 5 temporal attention heatmaps). Rather than briefly touching many modality pairs, we opted for a deep, well-justified analysis of one practically important pair. We will clarify this design choice and explicitly name other modality pairs as targets for future extensions. Also, our reverse transfer experiment from IMU to RGB in Table 5, indicates the superiority may be agnostic to other modalities and configurations.
>
> **Small/old datasets.** Our benchmark uses four public RGB–IMU HAR datasets—UTD-MHAD, CZU-MHAD, MMACT, and MMEA-CL—which were selected not for recency alone but for their diversity in scale, viewpoint (egocentric vs. third-person), environment, and sensor placement. These datasets span a range of ages and sizes: UTD-MHAD (2015, 861 samples) at the smaller end and MMACT (2021, a challenging multimodal dataset with multiple scenes, viewpoints, occlusions, and temporal variability) and MMEA-CL (2024, a large inthe-wild dataset of >6000 samples) at the more recent and larger end. Using older established datasets also brings the advantages of being well-documented, widely tested, and easy to compare against related work. Rather than introducing a new proprietary dataset, we standardize UMA splits (separate DHAR, DAlign, DVal, DTest) on these existing benchmarks and will release them for reproducibility, so that future work can plug in newer or larger datasets under the same protocol. We believe that the breadth of datasets and the depth of our analyses (including robustness, scaling, and visualizations) provide meaningful and generalizable insights, and they offer a solid foundation for future work to further extend and scale our methods.

---

> ### Author Response · Authors · 2025-12-03
> **Response to bgCj Part 4**
>
> ### **3. Analytical Depth**:
> **On Theory.** Our work is primarily an empirical contribution but is guided by a clear intuition: aligning local temporal features improves robustness to temporal distortions (shift, dilation, misalignment) that are common in continuous sensor data. This intuition is:
> - Illustrated in Figures 1–2 (corruption outside of a temporal latent time step’s input temporal receptive fields will not effect it in the UMA setup)
> - Quantitatively supported by robustness experiments (Table 2), where C3T degrades far less than ST and CA under temporal corruptions, and
> - Qualitatively supported by attention and t-SNE visualizations (Figures 6 and Appendix visualizations e.g., Figure 8), which show structured multimodal clusters and temporally-shifted attention patterns.
>
> We do not currently provide formal invariance bounds of LC3T. In the revision, we would be happy to provide a simple derived Lipschitz bound. We will state that LC3T is theoretically motivated but empirically validated, and we will flag a larger formal robustness analysis as a concrete avenue for follow-up work.
>
> Here is a simple derived bound we can add in the main paper (more details provided below):
> Under standard Lipschitz assumptions on the IMU encoder and HAR head, any temporal corruption whose input norm is below $\gamma / (2 L_f L_h)$ leaves the predicted label unchanged, where $\gamma$ is the margin between the correct class logit and others on clean data. In other words, C3T is provably invariant to a ball of temporal perturbations around each training trajectory, with radius inversely proportional to the Lipschitz constants and directly proportional to the classification margin.
>
>
> **On privacy and ethics.** This discussion is mentioned near the end of limitations in Section 6.4 in the appendix as it is less relevant to the technical contributions of this work, however, we thank the reviewer for making note and would be happy to introduce a short discussion of this in the main paper as well. IMU-based HAR is generally regarded as more privacy-preserving than video-based HAR, since IMU signals do not directly expose identities or rich visual details such as faces, and one motivation for UMA is precisely to transfer label-rich video models to privacy preserving sensor modalities. At the same time, cross-modal transfer can enable powerful inference from seemingly benign signals, which raises legitimate concerns about consent, potential misuse, and deployment practices. We will add a short subsection in the main paper that (i) explicitly acknowledges privacy and ethical considerations in sensor-based HAR and cross-modal transfer, (ii) notes that our experiments rely on public datasets collected under established consent procedures, and (iii) highlights practical mitigations (e.g., on-device inference, limiting raw data retention, clear user consent) and the need for responsible deployment.
>
> **On broader impacts.** The broader impacts of UMA-based RGB→IMU transfer include unlocking more advanced AI capabilities for a range of sensor modalities: models trained on well-labeled video can be transferred to modalities such as IMU, WiFi, or EMG for tasks like HAR, person identification, and gait analysis. This can support applications in daily health and biometric monitoring, improved smart appliances and home technologies, new interaction interfaces in XR and immersive environments, and more capable sensing in robotics. These potential applications are briefly mentioned in the introduction (lines 117–120); in the revised version, we will add one or two sentences in the conclusion or discussion to more clearly emphasize these broader impacts while reiterating that such capabilities must be developed and used with appropriate safeguards.
>
> ##  **Main Takeaways:**
> Main takeaways
> In summary, we hope our clarifications convey that:
> - C3T targets a strict UMA setting (zero labels in the target modality) that is practically relevant yet underexplored, and formalizes this setting for RGB→IMU HAR with standardized splits and baselines.
> - The core novelty lies in temporal token-level cross-modal alignment via LC3T, which consistently improves accuracy and temporal robustness over UMA-capable baselines, and is conceptually distinct from existing global-alignment, COMODO-style, or CoMM-style methods.
> - The work provides a deep empirical and qualitative analysis—including robustness to temporal noise, scaling behavior, fusion and reverse transfer, and multimodal latent visualizations—that offers new insight into how contrastive alignment behaves for time-series sensors and why temporal alignment is necessary.
>
>
> ## References:
> [1] Baiyu Chen, et al. Comodo: Cross-modal video-to-IMU distillation for efficient egocentric human activity recognition. arXiv:2503.07259, 2025.​
>
> [2] Benoit Dufumier, et al. What to align in multimodal contrastive learning? In International Conference on Learning Representations (ICLR), 2025.

---

> ### Author Response · Authors · 2025-12-03
> **Lipschitz Invariance Bound**
>
> In response to the request for a theoretical justification of robustness, we will add a simple invariance bound in the Appendix. Assuming the IMU encoder $f_{\mathrm{IMU}}$ and HAR head $h$ are Lipschitz, we show that for any temporal perturbation $\tilde x$ of an IMU sequence $x$ with $\|x-\tilde x\|\le\varepsilon$,
>
> $$ \|h(f_{\mathrm{IMU}}(x)) - h(f_{\mathrm{IMU}}(\tilde x))\| \le L_h L_f \varepsilon.$$
>
> If the clean example has a classification margin $\gamma$, then any temporal perturbation with $\varepsilon < \gamma / (2 L_f L_h)$ provably leaves the predicted label unchanged. Thus $L_{\mathrm{C3T}}$ yields invariance to a ball of temporal distortions whose radius is determined by the encoder/head Lipschitz constants and the learned margin. The local temporal alignment in $L_{\mathrm{C3T}}$ and the attention-based aggregation reduce the effective Lipschitz behavior on realistic time warps, which is consistent with the strong robustness observed in our crop/misalign/dilate experiments (Table 2).

---

### Official Review · Reviewer_95oV · 2025-10-31

**Soundness:** 3
**Presentation:** 3
**Contribution:** 2
**Rating:** 4
**Confidence:** 4

**Summary:**

This paper presents a novel method (C3T) and tackles an important problem, namely Unsupervised Modality Adaptation for cross-modal human activity recognition.

**Strengths:**

1. The paper clearly defines and formalizes the "Unsupervised Modality Adaptation" (UMA) setting, which is a meaningful and practical contribution for real-world sensor-based HAR systems.

2. The core idea of C3T—aligning fine-grained temporal latent vectors instead of a single global vector, which is intuitively appealing for time-series data.

3. The results demonstrate that C3T consistently and significantly outperforms strong baselines (ST and CA) in the UMA setting.

**Weaknesses:**

1. The paper only compares C3T against two adapted baseline families (Student-Teacher and Contrastive Alignment). There is no comparison with recent, powerful, and published methods designed for cross-modal learning, missing modality problems, or domain adaptation for time-series data.The paper fails to situate C3T within the current landscape of SOTA cross-modal HAR or video-IMU fusion techniques like Ts2act or Vi2ACT, to name a few. A reviewer cannot be confident that C3T represents a true advancement without these comparisons.
2. The ST and CA baselines, while reasonable starting points, are not pushed to their limits. For instance, was the temperature parameter (
τ) in the contrastive loss optimized for the CA baseline? Were more advanced distillation techniques explored for the ST baseline? Weak baselines may weaken the perceived performance improvement of C3T.
3. Performance on larger, more challenging datasets like MMACT (32.4% top-1 accuracy) is still very low, indicating that the method struggles with high diversity and is far from solving the general UMA problem.

**Questions:**

1. The performance improvement is attributed to the temporal alignment loss (L C3T ). However, C3T also uses a different HAR decoder (self-attention) compared to CA (MLP). The ablation in Table 4 attempts to address this but is confusing. It shows that a C3T variant with an MLP head can achieve higher accuracy (70.5%) on clean data than the chosen class-token attention head (62.5%). This critically undermines the claim that the self-attention mechanism is a key component for performance. It suggests that the primary benefit comes from the temporal alignment strategy itself, and the chosen architecture might even be suboptimal. A clearer ablation is needed.

---

> ### Author Response · Authors · 2025-12-03
> **Response to 95oV Part 1**
>
> We thank reviewer 95oV for their thoughtful review and insights. They emphasize our work's meaningful novel contribution to the community in terms of both the UMA problem formulation (1.) and the core C3T solution (2.), as well as the strong results (3.)
>
> We hope to provide some clarifications on the experimental methods to strengthen the empirical validity and trustworthiness, and show that the results are a true reflection of the model’s superiority to existing methods.
>
> ## Response to Weaknesses
> ### 1. **Comparison with cross-modal / missing-modality / domain adaptation methods**
>
> Our work focuses on a strict UMA regime: the target modality (IMU) has no labels during training, and the model must perform test-time inference purely from cross-modal structure. In the introduction, we note that most existing cross-modal and missing-modality methods assume some labeled data in each modality, which makes their architectures not directly applicable to UMA as defined in this paper (lines 53, 80–82), and we elaborate on this in Appendix §6.1. We provide a few of the missing modalities works here we attempted to use as baselines but could not adapt to the UMA setting (footnote $\dagger$) Traditional sensor-based domain adaptation methods further differ in that they operate within a single modality, transferring across distributions in the same input space (e.g., different users, devices, or sensor placements) [1], rather than across modalities. For instance, recent IMU domain generalization methods (e.g., RelCon ICLR 2025 [2]) learn domain-invariant representations from large-scale IMU data but still require labeled target data to adapt to a specific downstream task, and it is non-trivial to use them for RGB→IMU UMA without introducing exactly the type of label- or feature-level alignment that our ST and CA baselines already instantiate. Similarly, data augmentation is a common domain adaptation technique ([3]); however, augmenting one modality is not sufficient for being able to transfer to another modality without alignment.
>
> The methods mentioned by the reviewer face similar constraints. Ts2ACT [4], for example, uses an auxiliary (images) to improve performance when labeled sensor data are scarce, but it still relies on labeled target data: at the bottom of page 10 the authors state that 1% labeled data are used to build their representation model, and in Table 2 they only evaluate down to a 1-shot regime, not true zero-shot in the target modality. Ts2ACT is more of an augmented training technique than a transfer technique. To operate in UMA, Ts2ACT would need to be extended with pseudo-labeling or explicit cross-modal alignment, again essentially moving into the same families as our ST and CA baselines. Our ST and CA baselines are implemented with the same backbone components as C3T to ensure a fair comparison; adding additional training techniques (e.g., heavy data augmentation, large-scale pretraining, TS2ACT’s CLIP image encoders as auxiliary information during training) are valuable extensions that can be applied both to CA and C3T. The goal of this work is to introduce a new fundamental idea (fine-grained temporal alignment) and demonstrate its benefits through a controlled and interpretable analysis, rather than focusing on pushing state-of-the-art benchmark performance via implementation-specific techniques. Moreover, Ts2ACT uses an image encoder rather than a temporal video encoder, which sidesteps the core focus of our work: robust transfer over temporally-structured video to time-series IMU.
>
> Vi2ACT [5] extends TS2ACT with generated videos; however, it remains in the same few-shot cross-modal setting with labeled information to perform inference, which again falls outside the strict UMA setup. In contrast, our main experiments target zero-shot transfer to a new modality, and we only explore few-shot regimes as an extension (Appendix §6.5, Fig. 13), not as the primary focus. Even if such methods could be adapted to UMA with non-trivial modifications, our work contributes additional, complementary value: a detailed analysis of temporal robustness (Table 2), scaling behavior with model size (Fig. 4), latent-space and attention visualizations (Figs. 5–6), and experiments showing that the same architecture can handle supervised RGB-only, sensor fusion, reverse transfer, and UMA (Table 5). To our knowledge, existing cross-modal HAR methods do not provide this level of systematic analysis in the strict zero-label target-modality setting, and no existing methods perform alignment across a fine-grained temporal latent space.
>
> We appreciate the additional references suggested by the reviewer and will cite them in the revised version, making the distinction between their assumptions and our UMA setting more explicit. We can also move the Related Works discussion (Appendix Section 6.1) up to the main paper to make the distinction between UMA and previous works more evident.

---

> ### Author Response · Authors · 2025-12-03
> **Response to 95oV Part 2**
>
> ### 2. **On the strength and tuning of ST and CA baselines**:
>
> This work is centered on a new conceptual setting (UMA) and a new temporal alignment idea (C3T), and the baselines were designed to be as fair and controlled as possible for isolating that idea. All three methods—ST, CA, and C3T—share comparable backbone architectures and similar parameter counts in the main configuration (≈140M parameters in Figure 4), and Table 3 shows that simply swapping in more complex encoder types does not close the gap to C3T. This was intentional: rather than heavily engineering baselines in ways that we do not also apply to C3T, we keep the encoders and overall architectures comparable so that differences can be attributed primarily to the UMA mechanism, not to model capacity or additional tricks.
>
> We agree that, given more engineering effort, it is likely that ST and CA performance could be further improved—for example, via larger models, more data, richer augmentations, or more extensive hyperparameter sweeps. However, the same is true for C3T. Figure 4 suggests that, as representation capacity is reduced, C3T remains substantially more accurate than CA at every latent size, indicating that the relative advantage of temporal alignment is robust to scaling rather than a fragile artifact of a particular configuration. Our expectation is that if all methods were scaled up in parallel, C3T’s absolute numbers would increase while preserving much of its margin.
>
> For hyperparameters, we applied the same standard basic tuning effort to all methods, performing grid searches over learning rate and contrastive temperature $\tau$ on a validation split. The best-performing $\tau$ for CA and C3T was $\tau$=1, which is consistent with our setting. $\tau$ was originally introduced in large-scale models such as CLIP to temper noisy image–text pairs in massive datasets, whereas in our moderate-scale, well-synchronized RGB–IMU setting, a stronger “sharpening” effect was not beneficial. We therefore retain $\tau$ in the formulation for compatibility with prior work and potential future scaling, but empirically find that $\tau$=1 is the most appropriate choice in our current proof-of-concept experiments with limited resources.
>
> In the final version, we will clarify these design choices, emphasize that ST and CA were tuned under the same protocol as C3T, and explicitly present the parameter counts and hyperparameter choices.
>
> ### 3. **Raw Performance is Low on Large Datasets**:
> We agree that the absolute performance on MMACT is still modest, but view this as reflecting the difficulty of the dataset and the strictness of the UMA setting rather than a weakness specific to C3T. MMACT is intentionally challenging, with multiple scenes, heavy occlusions, diverse camera viewpoints, and variable start/stop times. In Table 1, even the fully supervised RGB model using the same backbone achieves only 42.1% Top-1 accuracy, which indicates that the underlying task is hard even when labels are available in the source modality. Given that C3T learns from labeled RGB and must generalize to unlabeled IMU, it is expected that UMA accuracy will be lower than the supervised RGB upper bound, and its difference is not that much distinct from the other datasets.
>
> Within this challenging regime, C3T still represents a clear step forward. On MMACT, ST and CA achieve 17.6% and 24.5% Top-1 accuracy, respectively, compared to 2.9% for random guessing, showing that all three methods perform non-trivial UMA. C3T further improves to 32.4%, substantially narrowing the gap to the supervised RGB model despite having no IMU labels during training. Moreover, our UMA results are in line with prior video-to-IMU alignment-and-transfer works on similar large-scale datasets (e.g., Mmg-Ego4d reports 35\% [6], IMU2CLIP 23\% [7], and Imagebind 25\% [8]), reinforcing that this is an open and difficult problem rather than one where near-perfect performance is expected.
>
> We will clarify in the paper that (1) UMA on large, heterogeneous datasets like MMACT is far from solved; (2) C3T does not claim to solve UMA, but to make a significant, measurable advance over existing UMA-capable baselines.

---

> ### Author Response · Authors · 2025-12-03
> **Response to 95oV Part 3**
>
> ## Response to Questions:
> ### 1. **C3T Attention Head**:
> The reviewer makes a great point here, and we would like to reiterate and highlight their observation, as it aligns with our intention. They correctly note that our ablation shows the C3T architecture without self-attention (using an MLP head) still outperforms the other methods (CA and ST), indicating that “the primary benefit comes from the temporal alignment strategy itself.” The reviewer then states that this “critically undermines the claim that the self-attention mechanism is a key component for performance.” However, there is a misunderstanding here: we do not claim that self-attention is the key component. In fact, we explicitly state the opposite in lines 403–405: “C3T’s strength lies in temporal alignment, not its attention mechanism.”
>
> We believe this actually strengthens the novelty and impact of our contribution. Using temporal attention to improve performance has been explored in standard video feature extractors (e.g., ViViT, VideoSwin), but performing *alignment and transfer* over fine-grained temporal latent features across modalities has, to our knowledge, not been proposed or studied. Our results show that this temporal alignment mechanism—rather than the specific choice of a self-attention head—is what uniquely boosts performance in our UMA setting for time-series sensor data. Self-attention, however, does indeed boost robustness to temporal noise and was thus chosen as the architecture for our main experiments.
>
> ##  **Main Takeaways:**
>
> Overall, we hope the above clarifications convey why C3T is a valuable and timely contribution, and why our baselines were carefully designed to prove this. While we agree that careful scrutiny of claimed state-of-the-art results is essential, the strict UMA setting we consider is relatively underexplored, and truly comparable baselines out of the box are difficult to find. In this context, we would kindly ask that the broader set of contributions be considered as a whole: the formalization of UMA for RGB→IMU HAR, the temporal alignment architecture, the extensive robustness and scaling experiments, and the qualitative analyses and explainability of multimodal latent spaces and alignment dynamics over time. In line with the ICLR guidelines that emphasize “new, relevant, impactful knowledge” rather than SOTA numbers alone, we believe our work provides exactly this: a clear problem formulation, a concrete and effective method, and a set of analyses that shed new light on contrastive cross-modal alignment and temporal transfer for time-series sensors.

---

> ### Author Response · Authors · 2025-12-03
> **Response to 95oV References**
>
> [1] Abhi Kamboj and Minh N. Do. A Survey of IMU Based Cross-Modal Transfer Learning in Human Activity Recognition. In International Conference on Learning Representations (ICLR), 2024.​​
>
> [2] Doris Y. Tsao, Max Xu, Yubo Liao, et al. RelCon: Relative Contrastive Learning for a Motion Foundation Model from Wearable Accelerometry. In International Conference on Learning Representations (ICLR), 2025.​​
>
> [3] Zikun Lyu and Yuxin Wang. CrossHAR: Generalizing Cross-dataset Human Activity Recognition via Hierarchical Self-Supervised Pretraining. In Proceedings of the ACM on Interactive, Mobile, Wearable and Ubiquitous Technologies (IMWUT), vol. 8, no. 2, 2024.​​
>
> [4] Kang Xia, Wenzhong Li, and Sanglu Lu. TS2ACT: Time Series to Action with Cross-Modal Contrastive Learning for Few-shot Human Activity Recognition. In Proceedings of the ACM on Interactive, Mobile, Wearable and Ubiquitous Technologies (IMWUT), vol. 7, no. 4, Article 187, 2023.​​
>
> [5] Kang Xia, Wenzhong Li, Yimiao Shao, and Sanglu Lu. Vi2ACT: Video-enhanced Cross-modal Co-learning with Representation Conditional Discriminator for Few-shot Human Activity Recognition. In Proceedings of the 32nd ACM International Conference on Multimedia (ACM MM), pp. 1848–1855, 2024.​​
>
> [6] Xinyu Gong, Sreyas Mohan, Naina Dhingra, Jean-Charles Bazin, Yilei Li, Zhangyang Wang, and Rakesh Ranjan. MMG-Ego4D: Multimodal Generalization in Egocentric Action Recognition. In Proceedings of the IEEE/CVF Conference on Computer Vision and Pattern Recognition (CVPR), pp. 6481–6491, 2023.​​
>
> [7] Seungwhan Moon, Andrea Madotto, Zhaojiang Lin, Alireza Dirafzoon, Aparajita Saraf, Amy Bearman, and Babak Damavandi. IMU2CLIP: Multimodal Contrastive Learning for IMU Motion Sensors from Egocentric Videos and Text. arXiv preprint arXiv:2210.14395, 2022.​​
>
> [8] Rohit Girdhar, Alaaeldin El-Nouby, Zhuang Liu, Mannat Singh, Kalyan Vasudev Alwala, Armand Joulin, and Ishan Misra. ImageBind: One Embedding Space to Bind Them All. In Proceedings of the IEEE/CVF Conference on Computer Vision and Pattern Recognition (CVPR), pp. 15180–15190, 2023.​​
>
> $\dagger$ Examples of missing modality baselines considered that cannot be applied to UMA: [1] Garcia, N. C., Morerio, P., & Murino, V. (2018). Modality Distillation with Multiple Stream Networks for Action Recognition. In V. Ferrari, M. Hebert, C. Sminchisescu, & Y. Weiss (Eds.), Computer Vision – ECCV 2018 (Vol. 11212, pp. 106–121). Springer International Publishing. [2] Wang, Q., Zhan, L., Thompson, P., & Zhou, J. (2020). Multimodal Learning with Incomplete Modalities by Knowledge Distillation. Proceedings of the 26th ACM SIGKDD International Conference on Knowledge Discovery & Data Mining, 1828–1838. [3] M. A. Nugroho, S. Woo, S. Lee and C. Kim, "AHFu-Net: Align, Hallucinate, and Fuse Network for Missing Multimodal Action Recognition," 2023 IEEE International Conference on Visual Communications and Image Processing (VCIP), Jeju, Korea, Republic of, 2023. [4] Park, Y., Woo, S., Lee, S., Nugroho, M. A., & Kim, C. (2023). Cross-modal alignment and translation for missing modality action recognition. Computer Vision and Image Understanding, 236, 103805. [5] Yang, L., Huang, Y., Sugano, Y., & Sato, Y. (2022). Interact before align: Leveraging cross-modal knowledge for domain adaptive action recognition. In Proceedings of the IEEE/CVF conference on computer vision and pattern recognition (pp. 14722-14732).

---

### Official Review · Reviewer_iCut · 2025-10-31

**Soundness:** 2
**Presentation:** 2
**Contribution:** 2
**Rating:** 4
**Confidence:** 3

**Summary:**

This paper introduces Cross-modal Transfer Through Time (C3T), a novel method for Unsupervised Modality Adaptation (UMA) in human activity recognition. C3T aligns temporal latent vectors across modalities (e.g., RGB and IMU) to preserve fine-grained temporal information, outperforming existing Student-Teacher and Contrastive Alignment methods.

**Strengths:**

Extensive experiments across four diverse datasets validate the method’s superiority in both accuracy and robustness. Ablation studies and visualizations (e.g., t-SNE, attention maps) provide strong empirical support.

**Weaknesses:**

This paper applies Unsupervised Modality Adaptation (UMA) to human activity recognition (HAR) and proposes a Cross-modal Transfer Through Time (C3T) method. The authors demonstrate that C3T outperforms ST and CA on a self-designed benchmark under the UMA setting, but I have several major concerns:
1.UMA has already been studied in other tasks, and merely applying it to HAR does not constitute significant novelty;
2.The baselines are overly simple, including only ST and CA, lacking comparisons with more advanced methods;
3.The benchmark is self-designed, and basic UMA methods are not evaluated on it, lacking comparison with existing work, so the claimed SOTA results are not convincing;
4.The method itself shows limited novelty, and the overall contribution is insufficient to justify a full-length paper.

**Questions:**

Please refer to Weakness.

---

> ### Author Response · Authors · 2025-12-02
> **Response to iCut Part 1**
>
> We would like to thank reviewer iCut for their time in reviewing this work. They recognize the strong empirical results, including the extensive experiments, ablations, and visualizations, as well as the accuracy and robustness gains of our method.
>
> We found the weaknesses in this review to be somewhat high-level. For example, when the reviewer refers to “other tasks,” “more advanced methods,” or “basic UMA methods,” it is not clear which works or methods are being referenced (the reviewer discussion phase, which unexpectedly had to be canceled, may have been helpful here). Nonetheless, we will try our best to respond to each point.
>
> 1. **On the novelty of UMA and our setting**:
> In our work, we introduce Unsupervised Modality Adaptation (UMA) as cross-modal transfer, where a model is trained with labels in some source modality and must perform inference in a target modality that has no labels during training. This setting is particularly challenging for time-series sensors, where temporal alignment across modalities is non-trivial, and labels are much scarcer than in video. We explicitly motivate a realistic deployment scenario where such a transfer is useful in a smart home with wearables (Figure 2). We believe formalizing a cross-modal transfer setting with a concrete application scenario and introducing an architecture designed for improved alignment of temporal data does constitute significant novelty.
>
>    Within this UMA setting, we categorize prior work into two families that can be adapted to our constraints: student–teacher/knowledge distillation methods and contrastive alignment methods. Our proposed C3T differs from these by performing alignment at a fine-grained temporal level (Figure 1), aligning sets of temporal latent tokens rather than single collapsed vectors. This design yields substantially better UMA accuracy (Table 1) and markedly higher robustness to temporal distortions such as time shift, misalignment, and dilation (Table 2), which are common in real-world continuous sensor data and critical for practical deployment. To the best of our knowledge, no prior work considers (i) this strict UMA setting, (ii) fine-grained temporal alignment across modalities, and (iii) systematic robustness to multiple types of temporal noise. If there are specific works you believe already address this combination of problem formulation, transfer architecture, or robustness analysis, we would be grateful for pointers so we can clearly differentiate our contributions.
>
> 2. **On the request for more advanced baselines**:
> We agree that strong baselines are important. In our work, the “CA” baseline is not a simple toy model but an implementation aligned with state-of-the-art contrastive alignment architectures used in video–IMU and multimodal representation learning (e.g., ImageBind-, IMU2CLIP-, and mmEgo4D-style contrastive setups), instantiated under our UMA constraints. These methods all follow the same core idea: align global latent embeddings across modalities to perform transfer. Although these methods were originally designed for image–text and then directly applied to time-series sensors.
>
>    C3T demonstrates that explicitly modeling the temporal structure of time-series sensors during alignment and transfer—by aligning temporal latent tokens instead of collapsing them into a single global vector—provides substantial gains in both performance and robustness. We will clarify this connection to existing advanced contrastive methods more explicitly and make it clearer that our CA baseline closely mirrors these architectures, while C3T can be viewed as a principled temporal extension of this line of work.

---

> ### Author Response · Authors · 2025-12-02
> **Response to iCut Part 2**
>
> 3. **On breadth of baselines and benchmark design**:
> Because the UMA setting is strict (no labels in the target modality), there are relatively few existing methods that can be directly applied without weakening the problem or adding supervision. Most missing modality works require some labels in the target modality. Within this constraint, we include two modern, representative families of approaches—student–teacher and contrastive alignment—as well as standard sensor-fusion baselines adapted to UMA. The fusion-based methods perform poorly under UMA (Appendix Table 11), illustrating that naïve extensions of supervised or semi-supervised techniques are not sufficient in this regime.
> To compensate for the limited number of truly compatible baselines, we provide a broad and detailed empirical study: four diverse public RGB+IMU datasets, comparisons to fully supervised counterparts, robustness experiments under multiple temporal corruptions, architecture and head ablations, and qualitative analyses (t-SNE, attention maps) that explain why aligning in a fine-grained temporal latent space is effective. We believe this depth of analysis meaningfully supports the novelty and usefulness of C3T despite the inherent difficulty of assembling a large set of fully comparable UMA baselines.
>
>    Our paper does not introduce a new private or arbitrary benchmark. Instead, it leverages well-established existing public RGB–IMU HAR datasets and configures the data to mimic the UMA setting we care about. The training splits (labeled RGB only, plus unlabeled paired RGB and IMU) and the testing split (unlabeled IMU only) are deliberately designed to enforce cross-modal transfer with zero IMU labels, reflecting a realistic and practically important scenario that could unlock scalable AI models for a wide range of time-series sensor applications. These datasets are also diverse in terms of activities, number of users, and number of samples, and they include both third-person and first-person viewpoints with varying start times, occlusions, and other real-world variations, further underscoring the impact and applicability of our setting and method.
>
> 4. **On the overall contribution and temporal latent alignment**:
> Unsupervised modality adaptation is a setting we explicitly define and formalize for the RGB→IMU HAR context. We clearly identify which prior methods can be adapted to this setting (student–teacher/distillation and contrastive alignment) and also point out their limitations: they treat temporal data by extracting features and then collapsing the entire sequence into a single latent vector before alignment. In contrast, C3T performs transfer directly in a temporal latent space, aligning local temporal tokens across modalities and then using a shared temporal model for HAR.
>
>    We evaluate this in a tough configuration where the target modality has no labels during training and show large gains over existing methods in this regime. Furthermore, we conduct an in-depth analysis of temporal robustness, considering several realistic types of noise for action recognition—dilation (different speeds), shifting (partial visibility in time), and temporal misalignment across modalities—and show that C3T maintains strong performance across all of them, whereas conventional methods degrade severely. To our knowledge, this setting and architecture have not been explored before, and our experiments and analysis would be useful to those interested in performing alignment with sensor data.
>
>
> **Main takeaways and planned clarifications:**
>
> In summary, our contributions are:
>
> - A clear motivation and formulation for RGB→IMU HAR unsupervised cross-modal transfer with a testing and evaluation protocol across multiple public datasets, as well as a categorization of existing methods that fit this setting.
> - A temporally-aware cross-modal transfer method (C3T) that aligns temporal latent vectors rather than full video embeddings, leading to significantly better accuracy and robustness to realistic temporal distortions than existing methods.
> - A comprehensive empirical and qualitative study (including robustness, ablations, fusion baselines, and visualizations) that provides insight into how and why temporal latent alignment is necessary for cross-modal transfer in sensor-based HAR.
>
> In the camera-ready version, we will sharpen the related work discussion around UMA and advanced multimodal baselines, highlight the uniqueness of temporal latent alignment and robustness analysis, and foreground these main takeaways in the introduction and conclusion.

---

### Author Response · Authors · 2025-12-03
**Overview of Rebuttal**

We thank all reviewers and the area chairs for their careful evaluations and constructive feedback. Here we provide a brief overview of the rebuttal. The reviews collectively highlight two main axes of discussion:
(i) the novelty and contributions of C3T
(ii) the strength and scope of our experimental evaluation.

Different reviews emphasize different aspects—some reviewers explicitly list limited novelty or contributions as a weakness while viewing the empirical rigor as a strength, whereas others regard the UMA formulation and C3T as novel practical contributions, but express concerns about experimental validity.

We believe the reviewers' concerns can be resolved through clarifications in understanding without needing to modify the main contributions of the paper itself. Here, we provide a brief overview of our novelty and experiments, and under the reviews, we provide comments with detailed line-by-line responses to the reviewers.


## Novelty:
Our contributions are threefold:
1. We formalize Unsupervised Modality Adaptation (UMA) for Human Activity Recognition (HAR), i.e., cross-modal transfer where the source modality (RGB) is labeled and the target modality (IMU) is completely unlabeled during training, and we categorize existing methods that can operate under this constraint (student–teacher and contrastive alignment). While there are works that perform related forms of zero-label cross-modal transfer, to our knowledge, none formalize this setting, motivate it with a concrete application scenario, systematically categorize applicable methods, and study them as extensively and in as much depth as we do here.
2. We propose C3T, which performs cross-modal alignment over fine-grained temporal latent vectors from convolutional feature maps, rather than a single global video embedding. This temporal token-level alignment, combined with a shared temporal head, yields significantly higher accuracy and robustness to temporal distortions than existing global contrastive alignment or student–teacher methods. To our understanding, no methods we are aware of have performed alignment across temporal features of two time-series modalities to capture fine-grained temporal information, and we believe this constitutes an interesting and novel contribution.
3. We provide a comprehensive empirical and qualitative study—including temporal noise robustness, model-size scaling, fusion and reverse-transfer experiments, and multimodal latent-space visualizations—that clarifies when and why temporal latent alignment is necessary for cross-modal transfer in sensor-based HAR. To our knowledge, very few works visualize and interpret multimodal contrastive latent spaces during training as we do (e.g., Figure 6), and none systematically evaluate cross-modal transfer under realistic temporal corruptions such as cropping, temporal shifts, and time dilation.


## Scope and Experiments:
This work aims to provide an in-depth understanding and explainable analysis of *how and why* C3T works on RGB video to IMU sensor transfer for HAR. Within this scope, our experiments show:
1) Comprehensive evaluations on diverse datasets (UTD-MHAD, CZU-MHAD, MMACT, MMEA-CL) consistently favor C3T over existing techniques.
2) Analysis of latent dimensions size and model size implies real-world practical applicability and scalability of C3T (Figure 4).
3) Temporal noise experiments (Table 2) demonstrate that C3T is significantly more robust than existing methods to realistic temporal corruptions (cropping, shifts, and dilation), which is a primary challenge for continuous, real-time HAR that C3T was designed to overcome. Qualitative analysis of attention heatmaps of temporal latent vectors supports this, showing that C3T focuses on relevant segments even under temporal shifts (Figure 5).
4) Multimodal t-SNE visualizations reveal that meaningful cross-modal clusters form in the latent space even before labels are introduced (Fig. 6), supporting the effectiveness of contrastive methods in UMA.
5) Architecture ablations (Tables 3, 4) support that the key performance gains stem from aligning fine-grained temporal vectors, rather than from auxiliary architectural choices.
6) Additional experiments (Table 5) (i) compare different training curricula for C3T, (ii) show that the learned model can operate on IMU only, RGB only, or fused modalities, and (iii) demonstrate that the same framework can be used for reverse transfer (IMU→RGB), suggesting that the approach is not tied to a particular configuration or modalities.

---

### Note · Authors · 2026-03-18

I have read and agree with the venue's withdrawal policy on behalf of myself and my co-authors.

---

### Meta-Review · Area_Chair_vnTT · 2025-12-29

**Summary:**

This submission studies Unsupervised Modality Adaptation (UMA) for cross-modal human activity recognition, focusing on transferring from labeled RGB video to unlabeled IMU at test time. The proposed method, Cross-modal Transfer Through Time (C3T), performs fine-grained temporal token alignment across modalities (rather than aligning a single pooled/global embedding), and reports consistent gains over two UMA-compatible baseline families (student–teacher distillation and contrastive alignment), alongside robustness analyses under temporal corruptions and several ablations.

Reviewers generally find the core idea of C3T, token-level temporal alignment under a strict Unsupervised Modality Adaptation setting, to be reasonable and practically motivated, but they raise consistent concerns about positioning, comparability, and evidentiary strength rather than technical correctness. The main acceptance barrier is that novelty remains borderline, as the paper does not yet draw a sufficiently sharp distinction from existing cross-modal HAR, video–IMU learning, missing-modality recognition, and sequence contrastive learning, even after the rebuttal’s clarifications. Baseline coverage is viewed as the most serious weakness, with reviewers unconvinced that comparisons limited mainly to student–teacher and contrastive alignment methods establish clear progress over the broader cross-modal and time-series adaptation literature, despite arguments about UMA incompatibility. Additional concerns include whether baselines are sufficiently tuned, confusion in ablations that blur the roles of temporal alignment versus architectural choices, and low absolute performance on challenging datasets, which suggests the problem remains far from solved and calls for more cautious framing. Overall, reviewers see the work as a meaningful but incremental step.

**Reviewer Concerns:**

The main weaknesses raised consistently across reviewers are about positioning, comparability, and clarity of evidence rather than the core intuition.

First, novelty and related-work positioning remain borderline. Reviewers agree the setting is practical, but question whether the contribution is more than “applying UMA to HAR,” and whether token-level temporal alignment is sufficiently differentiated from recent multimodal/sequence contrastive learning and video–IMU literature. The rebuttal helps by clarifying that the paper’s intended novelty is the strict UMA regime (zero labels in the target modality) and token-level temporal alignment, but the manuscript still needs a sharper, more explicit contrast to adjacent paradigms (cross-modal HAR, missing-modality action recognition, video–IMU pretraining, and sequence CL).

Second, baseline coverage and “SOTA” credibility remain the biggest acceptance blocker. All three reviewers explicitly argue that comparing primarily against ST and CA is not enough to establish advancement over the broader cross-modal/time-series adaptation landscape. The authors argue many methods are not directly applicable under strict UMA without modifying assumptions, which is reasonable, but the paper must still convince readers by (i) clearly listing which methods were attempted and why they are incompatible, and/or (ii) adding at least a small set of stronger, well-motivated comparisons (even if adapted variants) that a skeptical reader would expect in this space.

Third, baseline strength/tuning and ablation clarity need improvement. Reviewer 95oV flags that the CA/ST baselines may not be pushed (e.g., temperature tuning, distillation variants) and highlights a confusing ablation where the MLP head appears better on clean accuracy than the chosen attention head. The rebuttal clarifies that the main gain is from temporal alignment (not the attention head), but the paper should restructure ablations to cleanly isolate: alignment loss vs. decoder/head choice vs. robustness trade-offs, and make the “why this head is used” story unambiguous.

Finally, absolute performance on challenging datasets (e.g., MMACT) is still low, suggesting the problem is far from solved. The authors contextualize this with the supervised RGB upper bound and show relative improvements, which is useful, but the paper should present the takeaway as “meaningful step forward in a hard setting” rather than implicitly overclaiming resolution of UMA.

**Reviewer Scores:**

Reviewer iCut: No change or slight increase in score

Rationale: Reviewer iCut acknowledged strong empirical results and explicitly stated they “would not mind if the paper is accepted,” but their core concerns about limited novelty and insufficient baseline breadth remain largely conceptual. While the rebuttal clarifies the UMA setting and better motivates temporal token alignment, it does not fundamentally alter the reviewer’s view that the contribution may be incremental relative to existing cross-modal work. At best, clearer positioning could slightly soften the stance, but a full score increase is unlikely.

Reviewer 95oV: perhaps from marginally below to borderline)

Rationale: Reviewer 95oV was generally positive about the problem formulation and core idea, with concerns focused on baseline strength, ablation clarity, and low absolute performance on large datasets. The rebuttal directly and convincingly addresses several of these points, especially clarifying that temporal alignment, not the attention head, is the main contribution, and contextualizing MMACT performance. This reviewer appears receptive to such clarifications and could plausibly raise their score slightly, though probably not to a confident accept without stronger baselines.

Reviewer bgCj: No change

Rationale: Reviewer bgCj consistently framed the work as incremental and raised broader concerns about novelty, scope, theoretical grounding, and assumptions (paired data, limited modalities). While the rebuttal carefully differentiates C3T from related methods and acknowledges limitations, these responses are unlikely to fully resolve the reviewer’s more structural concerns. As such, their overall assessment would likely remain unchanged.

---

### Decision · Program_Chairs · 2026-01-26

Reject